# Image Translation as Diffusion Visual Programmers

**Cheng Han**[1,2]* **& James C. Liang**[1]* **& Qifan Wang**[3] **& Majid Rabbani**[1] **& Sohail Dianat**[1]
**& Raghuveer Rao**[4] **& Ying Nian Wu**[5] **& Dongfang Liu**[1†]
Rochester Institute of Technology[1], University of Missouri - Kansas City[2], Meta AI[3],
DEVCOM Army Research Laboratory[4], University of California, Los Angeles[5]

## Abstract

We introduce the novel Diffusion Visual Programmer (DVP), a neuro-symbolic image translation framework. Our proposed DVP seamlessly embeds a condition-flexible diffusion model within the GPT architecture, orchestrating a coherent sequence of visual programs (*i.e.*, computer vision models) for various pro-symbolic steps, which span RoI identification, style transfer, and position manipulation, facilitating transparent and controllable image translation processes. Extensive experiments demonstrate DVP's remarkable performance, surpassing concurrent arts. This success can be attributed to several key features of DVP: First, DVP achieves condition-flexible translation via instance normalization, enabling the model to eliminate sensitivity caused by the manual guidance and optimally focus on textual descriptions for high-quality content generation. Second, the framework enhances in-context reasoning by deciphering intricate high-dimensional concepts in feature spaces into more accessible low-dimensional symbols (*e.g.*, [Prompt], [RoI object]), allowing for localized, context-free editing while maintaining overall coherence. Last but not least, DVP improves systemic controllability and explainability by offering explicit symbolic representations at each programming stage, empowering users to intuitively interpret and modify results. Our research marks a substantial step towards harmonizing artificial image translation processes with cognitive intelligence, promising broader applications. Our demo page is released at here.

## 1 Introduction

Concurrent state-of-the-art image translation methods predominantly follow the paradigm of connectionism (Garcez et al., 2022), setting their goal as answering *"what"* — they aim to translate the image from the source domain to the designated target domain with fidelity. These methods can be broadly categorized into *Generative Adversarial Network (GAN)-based* and *Diffusion-based* methods (see §2). Diffusion-based approaches usually demonstrate superior capabilities when compared to their GAN-based counterparts, especially in the domains such as image quality and coherence, training stability, and fine-grained control (Dhariwal & Nichol, 2021). As such, they are perceived to harbor more auspicious potential in image translation (Mokady et al., 2023).

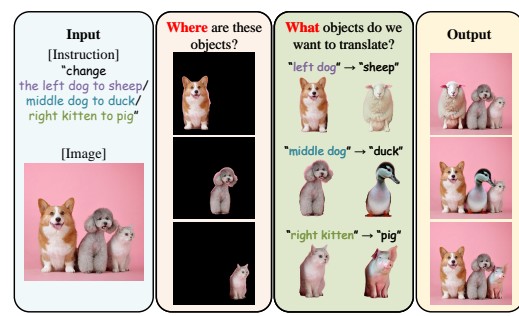

Figure 1: **Working pipeline showcase**. DVP represents a solution rooted in visual programming, demonstrating pronounced capabilities *in-context reasoning* and *explainable control*, in addition to its remarkable efficacy in style transfer.

---

*Equal contribution

†Corresponding author

Despite their huge success (Epstein et al., 2023; Dhariwal & Nichol, 2021), these diffusion-based methods exhibit a series of limitations: ① *Condition-rigid learning.* Existing arts following the principles in classifier-free guidance (Ho & Salimans, 2021; Mokady et al., 2023) face a significant challenge in effectively achieving a harmonious equilibrium between unconditional and conditional predictions. Typically, these methods rely on manually crafted guidance scale parameters to supervise the process of each individual image translation. This inherent restriction curtails the algorithmic scalability, thereby impeding their potential for achieving holistic automation for real-world applications; ② *Context-free incompetence.* Concurrent methods primarily engage in the global manipulation of diverse attributes (*e.g.*, stylistic and content-related elements) within images, which prioritize the maintenance of contextual integrity over local alterations. Nevertheless, the absence of context-free reasoning hinders the precision required to realize specific RoI modifications, while simultaneously upholding a broader sense coherence. Such contextual understanding capacities necessitate a high degree of semantic acumen and a robust mastery of image structure, which is sorely lacking within contemporary image translation solutions (Hitzler et al., 2022; Oltramari et al., 2020; Mao et al., 2019). ③ *System opacity.* Due to their black box nature, diffusion-based methods — coined by connectionism (Wang et al., 2023) — often exhibit a level of abstraction that distances them from the intrinsic physical characteristics of the problem they aim to model (Angelov & Soares, 2020). As a result, users encounter limited control over the model's behaviors before reaching the final outputs, rendering them rudderless in establishing trustworthiness in decision-making or pursuing systemic enhancement.

In this work, we revolutionize the focus of *"what"* into a more flexible image translation paradigm of *"where-then-what."* Under this paradigm, we first answer the question of *"where"* by finding the instructed target region accurately. After getting the target region (*i.e.*, Region of Interest (RoI)), we answers the question of *"what"* by translating it into the targeted domain with high-fidelity. Considering the aforementioned discussions, we are approaching the task of image translation from a fresh neuro-symbolic perspective, presenting a novel method named the Diffusion Visual Programmer (DVP). More concretely, we architect an condition-flexible diffusion model (Song et al., 2021), harmoniously integrated with the foundation of GPT (Brown et al., 2020), which orchestrates a concatenation of off-the-shelf computer vision models to yield the coveted outcomes. For example in Fig. 1, given an input instruction that specifies the translation from the "left dog" to a "sheep," DVP utilizes GPT as an AI agent to plan a sequence of programs with operations, subsequently invoked in the following procedure. It first addresses the fundamental question of *"where"* by identifying and segmenting the RoI of the "dog." Once segmented, the background undergoes an inpainting to restore the regions obscured by the "dog," and our condition-flexible diffusion model translates 🐕 → 🐑, addressing the query of *"what."* DVP, leveraging spatial data, positions the "sheep" in the instructed spatial location, and further enables various context-free manipulations (see §3.2).

DVP is an intuitive and potent image translation framework, showing superior performance qualitatively and quantitatively (§4.2) over state-of-the-art methods (Chen et al., 2023; Batzolis et al., 2021; Choi et al., 2021; Rombach et al., 2022; Saharia et al., 2022; Ruiz et al., 2023; Gal et al., 2022; Hertz et al., 2023; Mokady et al., 2023). It enjoys several attractive qualities: ❶ *Condition-flexible translation*. Our proposed condition-flexible diffusion model creatively utilizes instance normalization guidance (see §3.1) to mitigate the global noises in the unconditional embeddings by adaptive distribution shifting, and ensure that the model remains optimally conditioned based on the textual descriptions without any parameter engineering. It offers a streamlined solution tackling the challenge of hand-crafted guidance sensitivity on condition-rigid learning. This innovation addresses the *"what"* inquiry and enables a generalized learning paradigm for diffusion-based solutions. ❷ *Effective in-context reasoning*. By decoupling the high-dimensional, intricate concepts in feature spaces into low-dimensional, simple symbols (*e.g.*, [Prompt], [RoI object]), we enable context-free manipulation of imagery contents via visual programming (see §3.2). It essentially includes a sequence of operations (*e.g.*, segmentation, inpainting, translation) to establish in-context reasoning skills. Such a neuro-symbolic design fortifies our method with the capability to discern the concept of *"where"* with commendable precision. ❸ *Enhanced controllability and explainability*. Our modulating scheme uses explicit symbolic representations at each intermediate stage, permitting humans to intuitively interpret, comprehend, and modify. Leveraging by the step-by-step pipeline, we present not only a novel strong baseline but also a controllable and explainable framework to the community (see §4.3). Rather than necessitating the redesign of networks for additional functions or performance enhancement, we stand out for user-centric design, facilitating the seamless integration of future advanced modules.

## 2    RELATED WORK

**Image-to-Image Translation.**   Image-to-Image (I2I) translation aims at estimating a mapping of an image from a source domain to a target domain, while preserving the domain-invariant context from the input image (Tumanyan et al., 2023; Isola et al., 2017). For current data-driven methods, the I2I tasks are dominated into two groups: *GAN-based* and *Diffusion-based* methods. *GAN-based* methods, though they present high fidelity translation performance (Kim et al., 2022; Park et al., 2020a;b; Zhu et al., 2017a), pose significant challenges in training (Arjovsky et al., 2017; Gulrajani et al., 2017), and exhibit mode collapse in the output distribution (Metz et al., 2017; Ravuri & Vinyals, 2019). In addition, many such models face constraints in producing diverse translation outcomes (Li et al., 2023). *Diffusion-based* methods, on the other hand, have recently shown competitive performance on generating high-fidelity images. Works on conditional diffusion models (Choi et al., 2021; Rombach et al., 2022; Saharia et al., 2022) show promising performance, viewing conditional image generation through the integration of the reference image's encoded features during inversion (Mei et al., 2022). Though highly successful, they only offer coarse guidance on embedding spaces for generations (Chen et al., 2023), and exhibit ambiguity in intricate scenarios. In contrast, our approach, empowered by in-context reasoning, decouples complex scenes into low-dimensional concepts, and naturally handles such challenges.

**Text-guided Diffusion Models.**   Several concurrent works contribute to text-guided diffusion models. DreamBooth (Ruiz et al., 2023) and Textual Inversion (Gal et al., 2022) design pre-trained text-to-image diffusion models, while several images should be provided. Prompt-to-Prompt (Hertz et al., 2023; Mokady et al., 2023) manipulate local or global details solely by adjusting the text prompt. Through injecting internal cross-attention maps, they maintain the spatial configuration and geometry, thereby enabling the regeneration of an image while modifying it through prompt editing. However, (Hertz et al., 2023) does not apply an inversion technique, limiting it to synthesized images (Mokady et al., 2023). They both introduce extra hyper-parameter $w$ (*i.e.*, guidance scale parameter), which significantly influences the translated performance (see Fig. 5). In the light of this view, our approach re-considers the design of classifier-free guidance (Ho & Salimans, 2021) prediction, and further gets rid of such an oscillating parameter for robust prediction in a fully-automatic manner. We further include our intriguing findings in §4.3.

**Visual Programming.**   Visual programming serves as an intuitive way to articulate programmatic operations and data flows in addressing complex visual tasks (Gupta & Kembhavi, 2023), coined by the paradigm of neuro-symbolic. Concurrent visual programming, empowered by Large Language Models (LLMs), shows superior performance in various vision tasks (*e.g.*, visual relationship understanding (Donadello et al., 2017; Zhou et al., 2021), visual question answering (Yi et al., 2018; Mao et al., 2019; Amizadeh et al., 2020), commonsense reasoning (Arabshahi et al., 2021), image translation (Gupta & Kembhavi, 2023)), holding potential to change the landscape of image translation in terms of *transparency and explainability*, and *in-context reasoning*. We recognize a number of insightful approaches, where the most relevant ones to our work are AnyDoor (Chen et al., 2023), Inst-Inpaint (Yildirim et al., 2023) and VISPROG (Gupta & Kembhavi, 2023). Nevertheless, (Chen et al., 2023) select an object to a scene at a manually selected location, classifies objects by an additional ID extractor, and extract detail maps for hierarchical resolutions. (Yildirim et al., 2023) needs additional training on GQA-Inpaint dataset (Pfeiffer et al., 2022). Overall, these processes remain opaque or need additional training, which are not amenable to manual modifications or training-free paradigm. (Gupta & Kembhavi, 2023) introduces visual programming to compositional visual tasks. However, it mainly contributes to the paradigm of building a modular neuro-symbolic system, while the strong editing abilities for compositional generalization, has been overlooked. In addition, it simply replaces the original object by text-to-image generation, ignoring the preservation from any content from the original image. Our DVP, on the other hand, offers a nuanced framework that delineates each intermediate step in a manner that is both comprehensible and modifiable for human users (see §4.3). By translating complex feature spaces into lower-dimensional symbols, DVP enables fruitful context-free manipulations (see §3.2).

## 3    APPROACH

In this section, we introduce Diffusion Visual Programmer (DVP), a visual programming pipeline for image translation (see Fig. 2). Within our framework, image translation is decomposed into two distinct sub-objectives: ① style transfer, translating RoIs within images while upholding contextual coherence; and ② context-free editing, endowing the capacity for unrestricted yet judicious modifications. In response to ①, Condition-flexible diffusion model is introduced for autonomous,

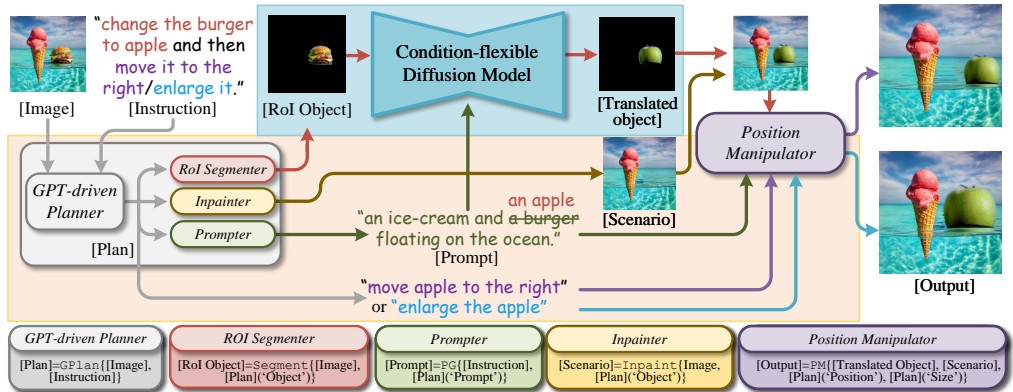

Figure 2: **Diffusion Visual Programmer (DVP) overview.** Our proposed framework contains two core modules: ▮ is the condition-flexible diffusion model (see §3.1), augmented by the integration of instance normalization (see Fig. 3), aimed to achieve a more generalized approach to translation; ▮ stands for visual programming (see §3.2), fulfilled by a series of off-the-shelf operations (*e.g.*, `Segment` operation for precise RoI segmentation). The overall neuro-symbolic design enables in-context reasoning for context-free editing. We also enjoy enhanced controllability and explainability by intuitively explicit symbols (*e.g.*, [Prompt], [RoI object], [Scenario], [Translated object]) at each intermediate stage, facilitating human interpretation, comprehension and modification.

non-human-intervened translation (§3.1). To achieve ②, we present In-context Visual Programming, which decomposes high-level concepts into human-understandable symbols, enabling adaptable manipulation (§3.2). We elaborate on our techniques below.

## 3.1 CONDITION-FLEXIBLE DIFFUSION MODEL

**Preliminaries.** Text-guided diffusion models transform a stochastic noise vector $z_t$ and a textual prompt embedding $\mathcal{P}$ into an output image $z_0$ aligned with the given conditioning prompt. For achieving step-by-step noise reduction, the model $\epsilon_\theta$ is calibrated to estimate synthetic noise as:

$$\min_\theta E_{z_0, \epsilon \sim N(0,I), t \sim \text{Uniform}(1,T)} ||\epsilon - \epsilon_\theta(z_t, t, \mathcal{P})||^2, \tag{1}$$

where $\mathcal{P}$ is the conditioning prompt embedding and $z_t$ is a hidden layer perturbed by noise, which is further introduced to the original sample data $z_0$ based on the timestamp $t$. During inference, the model progressively eliminates the noise over $T$ steps given a noise vector $z_T$. To accurately reconstruct a given real image, the deterministic diffusion model sampling (Song et al., 2021) is:

$$z_{t-1} = \sqrt{\frac{\alpha_{t-1}}{\alpha_t}} z_t + \left( \sqrt{\frac{1}{\alpha_{t-1}} - 1} - \sqrt{\frac{1}{\alpha_t} - 1} \right) \cdot \epsilon_\theta(z_t, t, \mathcal{P}), \tag{2}$$

where $\alpha_t := \prod_{i=1}^{t}(1 - \beta_i)$, and $\beta_i \in (0, 1)$ is a hyper-parameter for noise schedule. Motivated by (Hertz et al., 2023), we generate spatial attention maps corresponding to each textual token. Specifically, a cross-attention mechanism is incorporated for controllability during translation, facilitating interaction between the image and prompt during noise prediction as:

$$\epsilon_\theta(z_t, t, \mathcal{P}) = \text{Softmax}(\frac{\mathtt{l}_Q(\phi(z_t))\mathtt{l}_K(\psi(\mathcal{P}))}{\sqrt{d}})\mathtt{l}_V(\psi(\mathcal{P})), \tag{3}$$

where $\mathtt{l}_Q$, $\mathtt{l}_K$, $\mathtt{l}_V$ are learned linear projections. $\phi(z_t)$ represents spatial features of the noisy image, and $\psi(\mathcal{P})$ stands for the textual embedding.

**Instance Normalization Guidance.** Text-guided generation frequently faces the challenge of magnifying the influence of the conditioning text during the generation process (Song et al., 2021). (Ho & Salimans, 2021) thus put forth a classifier-free guidance approach, making an initial prediction generated without any specific conditioning. This unconditioned output with the guidance scale parameter $w$ is then linearly combined with predictions influenced by the conditioning text with scale $(1 - w)$. Formally, given $\varnothing = \psi(\cdot)$ as the feature representation from the null text, we have:

$$\tilde{\epsilon}_\theta(z_t, t, \mathcal{P}, \varnothing) = w \cdot \epsilon_\theta(z_t, t, \mathcal{P}) + (1 - w) \cdot \epsilon_\theta(z_t, t, \varnothing). \tag{4}$$

In practice, we observe that the scaling factor $w$ is highly sensitive. Even minor fluctuations in its value can lead to substantial effects on the final images. This stringent requirement for fine-tuning on a per-image basis makes its widespread adoption in real-world applications impractical. In the light of this view, we introduce the concept of adaptive distribution shift for the condition-flexible translation. Specifically, we carefully consider the role of the two types of embeddings (*i.e.*, the conditional prediction, $\epsilon_\theta(z_t, t, \mathcal{P})$ and the unconditional noise prediction $\epsilon_\theta(z_t, t, \varnothing)$ ). We argue that these embeddings warrant differentiated treatment, transcending a rudimentary linear combination from Eq. 4. In the works of (Ulyanov et al., 2016; 2017), it is posited that the efficacy of instance normalization in style transfer can be attributed to its robustness against fluctuations in the content image. Classifier-free guidance (Ho & Salimans, 2021) exhibits a similar form, which blends stable predictions from a conditional embedding with those concurrently trained unconditional embedding.

We, therefore, utilize instance normalization and find that it effectively aligns with the intended translation direction by reducing the impact of unconditional embeddings (see Fig. 3). This ensures the diffusion model remains exclusively conditioned on the prompt, eliminating any undesired variations. The introduction of instance normalization not only enhances the translation performance but also fortifies the model's ability to handle variations from the input distribution, regardless of potential differences in unconditional distributions (see Fig. 5). Formally, the instance normalization guidance is formulated as:

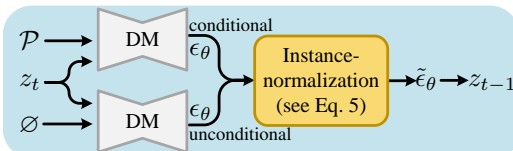

Figure 3: **Instance Normalization Guidance.**

$$\tilde{\epsilon}_\theta(z_t, t, \mathcal{P}, \varnothing) = \sigma(\epsilon_\theta(z_t, t, \varnothing))conv(\frac{\epsilon_\theta(z_t, t, \varnothing) - \mu(\epsilon_\theta(z_t, t, \mathcal{P}))}{\sigma(\epsilon_\theta(z_t, t, \mathcal{P}))}) + \mu(\epsilon_\theta(z_t, t, \varnothing)), \quad (5)$$

where $conv$ represents a $1 \times 1$ conv layer. Intuitively, the network $\tilde{\epsilon}_\theta$ is re-scaled with $\sigma$ and shift it with $\mu$, where $\mu$ and $\sigma$ represent the mean of the unconditional embedding and the standard deviation of the conditional embedding, respectively. Given that both $\mu$ and $\sigma$ are known values from predictions, we elegantly abstain from operations that are influenced by human intervention for further unconditional textual embedding tuning. Ablative studies in §4.3 further demonstrate our contributions.

### 3.2    IN-CONTEXT VISUAL PROGRAMMING

Condition-flexible diffusion model in §3.1 provides a generalized image translation solution, addressing the *"what"* inquiry as the neural embodiment part in our proposed framework. We further present in-context visual programming, using a symbolic reasoning process to bridge the learning of visual concepts and text instructions, understanding the concept of *"where"* (see Fig. 6). By breaking down the rich complexity of high-dimensional concepts into simpler, low-dimensional symbolic forms, we set the stage for a cascade of logical operations. This, in turn, enables the development of nuanced, in-context reasoning capabilities. The core components are articulated below.

**Symbols.**  Within our framework, we produce intermediate steps such as [Prompt], [RoI object], [Scenario]. These intermediary outcomes not only facilitate transparency and explainability for human interpretation, comprehension, and modifications, but can also be considered as symbols (Cornelio et al., 2022). These symbols serve as low-dimensional data enable structured operations, bridging the gap between raw data-driven methods and symbolic reasoning.

**Operations.**  We carefully grounded five operations, {GPlan, PG, Segment, Inpaint, PM}, in in-context visual programming (see Fig. 2). We first leverage the power of large language model — GPT-4 (OpenAI, 2023) as an AI agent to perceive, plan and generate program directives (Jennings, 2000; Anderson et al., 2018). This component, named as the *GPT-driven Planner* module, manages corresponding programs with few examples of similar instructions, and invokes requisite operations via the GPlan operation in the reasoning process. All programs adhere to the first-order logic principles to identify attributes, allowing us to express more complex statements than what can be expressed with propositional logic alone (Mao et al., 2019). Specifically, these programs have a hierarchical structure of symbolic, functional operations (*i.e.*, PG, Segment, Inpaint, PM), guiding the collections of modules (*i.e.*, *Prompter*, *RoI Segmenter*, *Inpainter* and *Position Manipulator*) for each phase. Note that the operations can be executed in parallel, offering flexible combinations for systemic controllability via the program execution (see Fig. 7). Specifically, *Prompter* utilizes GPT-4 to generate detailed image descriptions on any given input image by the operation PG. Our input image therefore is not restricted solely to human-annotated images as in previous arts (Mokady et al., 2023; Ho & Salimans, 2021; Chen et al., 2023); rather, it enables random unlabeled images as

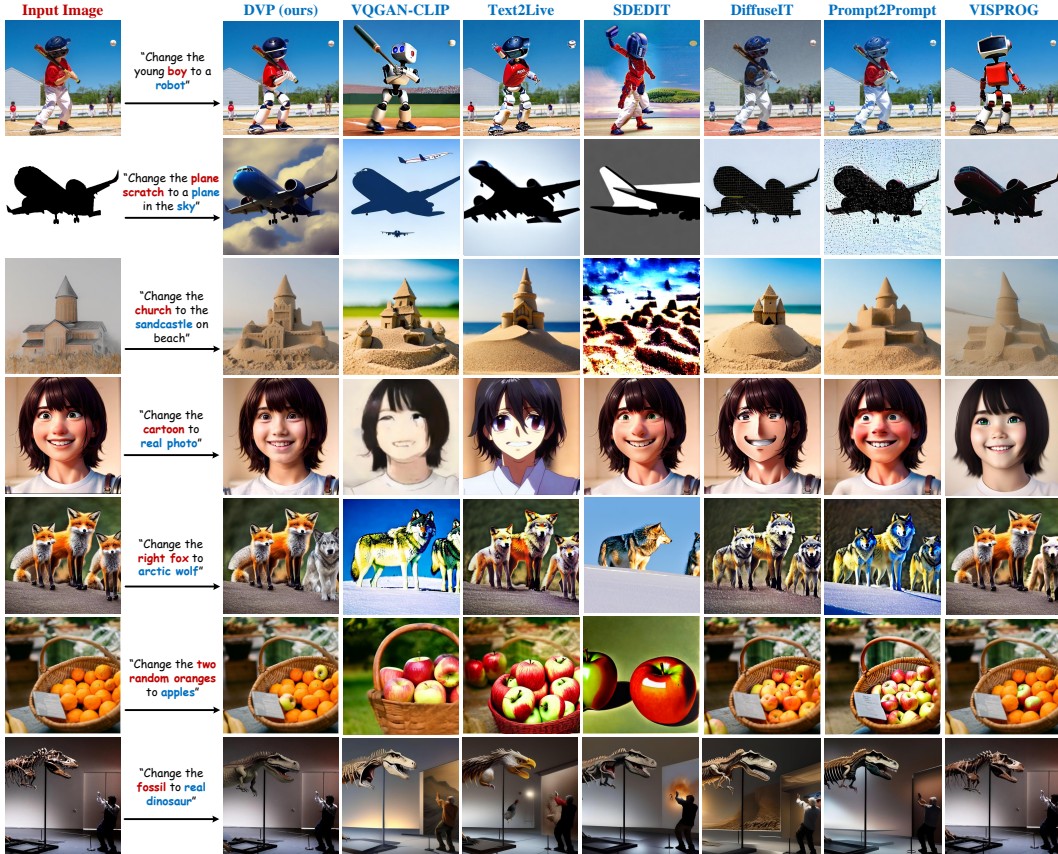

Figure 4: **Qualitative results with the state-of-the-art baselines.** DVP exhibits rich capability in style transfer, achieving realistic quality while retaining high fidelity. Owing to the context-free manipulation (see §3.2), the DVP framework is capable of flawlessly preserving the background scenes while specifically targeting the translation of the RoI. Note that while VISPROG also enables context-free editing, it exhibits considerable limitations in rational manipulation (see Fig. 6).

inputs, thereby broadening the potential application scenarios and augmenting data efficiency. *RoI Segmenter* creates flexible RoI segmentation via a pre-trained Mask2former (Cheng et al., 2022), in alignment with the operation of `Segment`. *Inpainter* operates based on the logic of ¬RoI, indicating the completion of foreground/background via stable diffusion v1.5 (Rombach et al., 2022; Lugmayr et al., 2022), assigned to the `Inpaint` operation. We further classify rational concepts (*e.g.*, position, scale) by *Position Manipulator* (*i.e.*, assigned as the `PM` operation), translating the programmed human language instructions into a domain-specific language (DSL) designed for positioning. The DSL covers a set of fundamental logic such as `Enlarge`, `Shrink`, `Left`, `Right`, `Up`, and `Down` *etc.* These intuitive logic share the same input format and output interface, thus achieving programming flexibility (see Fig. 7).

**Program Execution.** The programs are managed by the *Compiler* (Gupta & Kembhavi, 2023), which creates a mapping association between variables and values. It further steps through the program by incorporating the correct operations line-by-line. At every stage of execution, the program will activate the specified operation, and the intermediate outputs (*e.g.*, [prompts], [RoI object]) are in human-interpretable symbols, thereby enhancing the explainability of the system. Since these outputs enable direct visual evaluations, offering an option to either repeat the current step or proceed to the next one, which in turn improves the system's controllability, demonstrated in §4.3.

# 4 EXPERIMENTS

## 4.1 IMPLEMENTATION DETAILS

**Benchmarks.** For quantitative and qualitative results, we conduct a new benchmark (see §E), consisting of 100 diverse text-image pairs. Specifically, we manually pick images from web, generated

images, ImageNet-R (Hendrycks et al., 2021), ImageNet (Russakovsky et al., 2015), MS COCO (Lin et al., 2014), and other previous work (Ruiz et al., 2023; Tumanyan et al., 2023). To ensure a fair comparison and analysis, we construct a list of text templates including 20 different classes for translating, each containing five images of high-resolution and quality. For each originating class (*e.g.*, building), we stipulate corresponding target categories and stylistic attributes , thereby facilitating the automated sampling of various permutations and combinations for translating and evaluation.

**Evaluation Metrics.** We follow (Ruiz et al., 2023; Chen et al., 2023), and calculate the CLIP-Score (Hessel et al., 2021) and DINO-Score (Caron et al., 2021). These metrics enable the similarity between the generated image and the target prompt. We further conduct user studies involving 50 participants to evaluate 100 sets of results from multiple viewpoints, such as fidelity, quality, and diversity. Each set comprises a source image along with its corresponding translated image. To facilitate the evaluation, we establish comprehensive guidelines and scoring templates, where scores range from 1 (worst) to 5 (best). Please refer to §M for more details.

## 4.2 COMPARISONS WITH CURRENT METHODS

**Qualitative Results.** For fair comparison, we include six state-of-the-art baselines (*i.e.*, SDEdit (Meng et al., 2022), Prompt-to-Prompt (Hertz et al., 2023), DiffuseIT (Kwon & Ye, 2023), VQGAN-CLIP (Crowson et al., 2022), Text2LIVE (Bar-Tal et al., 2022) and VISPROG (Gupta & Kembhavi, 2023) on diverse prompt-guided image translation tasks. We present the visualization results with DVP in Fig. 4. For style transfer, incorporating instance normalization imbues our translated images to strike a balance between high preservation to the guidance layout, and high fidelity to the target prompt. For example, DVP vividly transfers the "young boy", "church", *etc*. into "robot", "sandcastle", *etc*. in Fig. 4. Other methods contrastingly yields corrupted translation on these RoI or introduce excessive variations. DVP also exhibits strong in-context reasoning by acknowledging and transferring objects as instructed by the user. For example, DVP selectively modify the "right fox" to the "arctic wolf," while leaving other foxes and the background unaffected (see Fig. 4 fifth row). Though VISPROG enables local editing, it falls short in cultivating the context reasoning skills, and result in translating all "foxes" into "arctic wolves" (see §2). The overall results demonstrate that SDEdit and VQGAN-CLIP compromises between preserving structural integrity and allowing for significant visual changes. DiffuseIT and Prompt2Prompt maintain the shape of the guidance image but makes minimal alterations to its appearance. Text2Live exhibits associations between text and certain visual elements (*e.g.*, the shape of objects); however, translating the objects to entirely new ones may not lead to visually pleasing results. In contrast, DVP effectively handles all examples across various scenarios and instructions.

**Quantitative Comparisons.** Discussed in §4.1, we further report the user study, CLIP-Score, and DINO-Score in Table 1. The results are consistent with our visual evidence: DVP distinctively outperforms other competitors in both fidelity and quality, most notably in the realm of fidelity. While existing methods are designed to consider merely on the semantic consistency, DVP goes one step further by introducing visual programming understanding with cognitive reasoning that preserving instance identity and background scenarios. This inherently

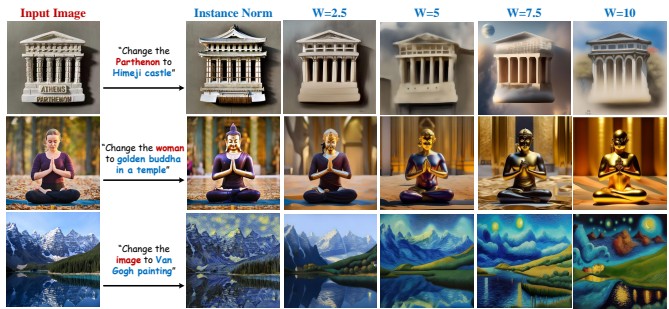

Figure 5: Ablative visualization **results of instance normalization** compared with various guidance scales $w$.

allows for superior performance in terms of fidelity, thereby making our model more versatile and applicable in various scenarios.

## 4.3 SYSTEMIC DIAGNOSIS

This section ablates our core design of DVP on *instance normalization guidance* in translation (§3.1) and *in-context reasoning* in visual programming (§3.2). Our systemic merits in *explainable controllability* as well as *label efficiency*, coined by the neuro-symbolic paradigm, are also discussed.

Table 1: **User study, CLIP-Score, and DINO-Score** on the comparison between our proposed framework and state-of-the-art baselines. The metrics are detailed in §4.1.

| Method | User Study | | | Quantitative Results | |
|---|---|---|---|---|---|
| | Quality | Fidelity | Diversity | CLIP-Score | DINO-Score |
| VQGAN-CLIP (Crowson et al., 2022) | 3.25 | 3.16 | 3.29 | 0.749 | 0.667 |
| Text2Live (Bar-Tal et al., 2022) | 3.55 | 3.45 | **3.73** | 0.785 | 0.659 |
| SDEDIT (Meng et al., 2022) | 3.37 | 3.46 | 3.32 | 0.754 | 0.642 |
| Prompt2Prompt (Hertz et al., 2023) | 3.82 | 3.92 | 3.48 | 0.825 | 0.657 |
| DiffuseIT (Kwon & Ye, 2023) | 3.88 | 3.87 | 3.57 | 0.804 | 0.648 |
| VISPROG (Gupta & Kembhavi, 2023) | 3.86 | 4.04 | 3.44 | 0.813 | 0.651 |
| DVP (ours) | **3.95** | **4.28** | 3.56 | **0.839** | **0.697** |

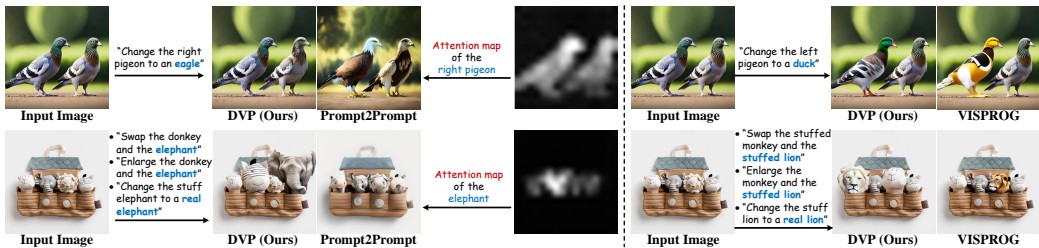

Figure 6: Visualization **results of in-context reasoning** against attention-based Prompt2Prompt (Mokady et al., 2023) and programming-based VISPROG (Gupta & Kembhavi, 2023).

**Instance Normalization.** Our condition-flexible diffusion model diverges from conventional approaches (Mokady et al., 2023; Ho & Salimans, 2021) by employing instance normalization guidance to enhance the robustness of translations and capacity to manage variations in the input distribution (see §3.1). To verify diffusion model methods with instance normalization and guidance scale (Mokady et al., 2023), we conduct extensive experiments in Fig. 5 and Table 2(a) qualitatively and quantitatively. For fairness, these comparisons are made *without* incorporating in-context visual programming into our approach. We opt for the guidance scale parameter $w$ range centered around 7.5 for linear combination, which is chosen based on the default parameter provided in (Mokady et al., 2023; Ho & Salimans, 2021). The parameter is varied with a step size of 2.5 to investigate its influence on the translated images. While the guidance scale baseline exhibits substantial variations on translated images (see Fig. 5 right 4 columns) when $w \in \{2.5, 5, 7.5, 10\}$, instance normalization enables high fidelity with natural, stable translation *without any* manually tuned parameter.

**In-context Reasoning.** DVP employs a set of visual programming operations for image translation, thereby facilitating a powerful in-context reasoning capability during image manipulation. To better demonstrate our claim, we present and compare our translated results with strong cross-attention map baseline method Prompt2Prompt (Mokady et al., 2023) in Fig. 6. Specifically, in the first row of Fig. 6, we aim to translate *only* the right pigeon to an eagle. The cross-attention map on Prompt2Prompt indicates that it recognizes both pigeons, albeit with a notable failure to discern the positional information accurately. Consequently, the model erroneously translates both pigeons into eagles. In contrast, our approach exhibits a strong in-context understanding of the

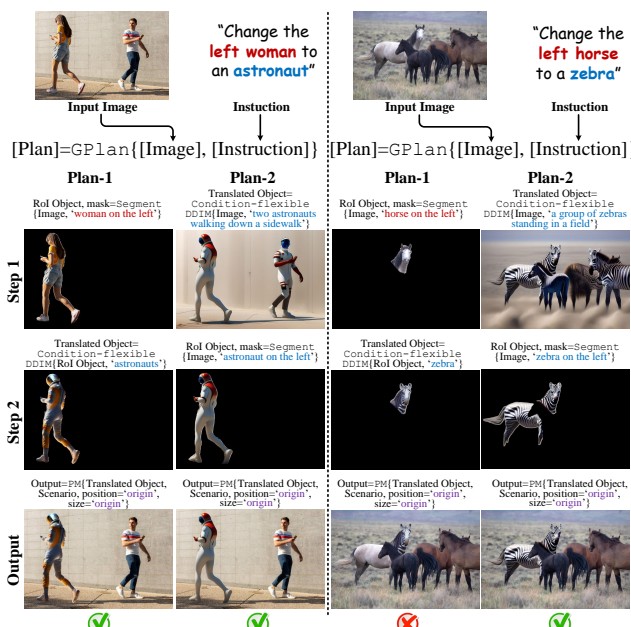

Figure 7: **Explainable controllability** during program execution, which is easy for error assessment and correction.

Table 2: **Ablative results** on instance normalization guidance and label efficiency.

| Metrics | $w = 2.5$ | $w = 5$ | $w = 7.5$ | $w = 10$ | **Inst. norm** | Methods | CLIP-Score | DINO-Score |
|---|---|---|---|---|---|---|---|---|
| CLIP-Score | 0.833 | 0.795 | 0.742 | 0.686 | **0.839** | Human-annotated | 0.817 | 0.688 |
| DINO-Score | 0.664 | 0.681 | **0.712** | 0.678 | 0.697 | *Prompter* | **0.839** | **0.697** |

|  (a) Instance normalization  |  (b) Label efficiency  |

scene and accurately translates the designated pigeon as instructed. We design more sophisticated instructions in the second row example in Fig. 6. For Prompt2Prompt, the ambiguous cross-attention on elephant results in wrongful artifacts across various animals in translation. It also demonstrates limitations in performing RoI relation editing (*i.e.*, Swap, Enlarge). We further compare DVP with VISPROG (Gupta & Kembhavi, 2023), a visual programming approach that enables local image editing (see §2). As seen, although both DVP and VISPROG allow for instructive object replacement (Fig. 6 right section), VISPROG is inferior in translation fidelity (*e.g.*, translated duck is too yellow and lion has a different pose). VISPROG also fails to support relation manipulations while DVP exhibits a commendable result in all examples following strictly to human instructions.

**Explainable Controllability.** We next study the explainable controllability during program execution (see §3.2). In design, we enable multiple operations worked in parallel, there are different program plans available for a diverse order of operation sequences. As shown in Fig. 7 (left section), "change the left woman to an astronaut" can be achieved by using both **Plan-1** and **Plan-2**. Throughout the execution process, the program is run line-by-line, triggering the specified operation and yielding human-interpretable intermediate outputs at each step, thereby facilitating systemic explainability for error correction. As shown in Fig. 7 (right section), **Plan-1** mistakenly segments only the head of the "left horse," leading to an incomplete translation to "zebra." Owing to the provision of explainable output at each step, we are able to identify specific issues and subsequently employ alternative **Plan-2** as a more suitable strategy for translating the "left horse."

**Label Efficiency.** *Prompter* generates detailed image descriptions for arbitrary input images, thereby relaxing label dependency without being tightly bound by human annotations (see §3.2). To verify its efficacy, we introduce user study (*i.e.*, Table 2(b)) and further provide visual evidences (*i.e.*, Fig. 8), contrasting human-generated annotations with those produced by *Prompter*. Given that the focus of this ablative study is to investigate the impact of labels on translated images, visual programming is

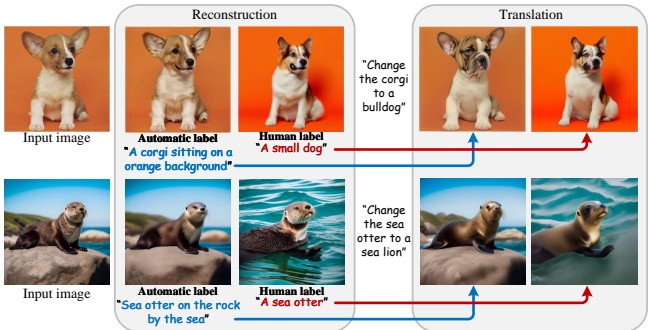

Figure 8: **Human-annotated and *Prompter*-generated descriptions.** Descriptions generated via the *Prompter* yield finer reconstructions, as well as subsequent translated outputs, suggesting a relaxation of label dependency.

excluded from the experimental design. In Fig. 8, we bypassed 60% of the optimization steps. Otherwise, the reconstructed images would closely resemble the input images, irrespective of the prompts, even in scenarios where the prompts are entirely erroneous. As seen, the GPT-4 generated annotations result in superior performance in both CLIP-Score and DINO-Score (*i.e.*, Table 2(b)), indicating that the generated annotations benefit the quality of the translated images. The visual evidences in Fig. 8 support our claim: by having detailed descriptions of images, the reconstruction phase yields finer results, which subsequently manifest in the quality of the final translated images.

## 5 CONCLUSION

In this work, we introduce DVP, a neuro-symbolic framework for image translation. Compared to concurrent image translation approaches, DVP has merits in: **i)** generalized translation without considering hand-crafted guidance scales on condition-rigid learning; **ii)** simple yet powerful in-context reasoning via visual programming; **iii)** intuitive controllability and explainability by step-by-step program execution and parallel operations. As a whole, we conclude that the outcomes presented in our paper contribute foundational insights into both image translation and neuro-symbolic domains.

## ACKNOWLEDGMENT

This research was supported by the National Science Foundation under Grant No. 2242243 and the DEVCOM Army Research Laboratory under Contract W911QX-21-D-0001.

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

## SUMMARY OF THE APPENDIX

This appendix contains additional details for the ICLR 2024 submission, titled *"Image Translation as Diffusion Visual Programmers."* The appendix is organized as follows:

- §A Implementation details and the pseudo-code.
- §B More quantitative results for instance normalization.
- §C More quantitative results for in-context reasoning.
- §D Video translation.
- §E General-context dataset.
- §F Runtime analysis.
- §G More qualitative results.
- §H Module Replacement Analysis.
- §I More comparison with Instruct-Pix2Pix.
- §J Ambiguous instruction analysis.
- §K Complex instruction analysis.
- §L Error analysis.
- §M Introduction of user study metrics and more user study results.
- §N Discussion of limitations, societal impact, and directions of our future work.

## A  IMPLEMENTATION DETAILS AND PSEUDO-CODE OF DVP

**Hyper-parameters.** We follow the common practices (Ruiz et al., 2023; Hertz et al., 2023; Kwon & Ye, 2023) and use stable diffusion v1.5 (Rombach et al., 2022) as the base generator. For null-text optimization (Mokady et al., 2023), we process the image resolution to $512 \times 512$. We choose AdamW (Loshchilov & Hutter, 2019) optimizer with an initial learning rate of 1e-5, betas = (0.9, 0.999), eps = 1e-8 and weight decays of 0.01 as default.

**Condition-flexible Diffusion Model.** For all experiments, we utilize diffusion model (Song et al., 2021) deterministic sampling with 50 steps. The deterministic diffusion model inversion is performed with 1000 forward steps and 1000 backward steps. Our translation results are performed with 50 sampling steps, thus we extract features only at these steps. Within each step, null-text optimization (Mokady et al., 2023) is employed for inference time pivotal fine-tuning *without* additional training schedule. Specifically for each input image, our approach optimizes the $1 \times 1$ conv layer and the unconditional embedding concurrently. Furthermore, we follow the common practise (Mokady et al., 2023; Hertz et al., 2023) and re-weight the positive and negative prompting via the hyper-parameter, which is used to magnify/depreciate the attention during the conditional prediction and unconditional prediction process.

**In-context Visual Programming.** We choose the GPT-4 (OpenAI, 2023) as our *Planner* discussed in §3.2, utilizing the official OpenAI Python API. The maximum length for generated programs is set to 256, and the temperature is set to 0 for the most deterministic generation. Furthermore, we utilize Mask2Former (Cheng et al., 2022) as a segmenter, Repaint (Lugmayr et al., 2022) as an inpainter, and BLIP (Li et al., 2022) as a prompter.

**Competitors.** We employ the official implementations provided for each competitor for fair comparisons: VISPROG (Gupta & Kembhavi, 2023), Prompt-to-Prompt (Hertz et al., 2023), DiffuseIT (Kwon & Ye, 2023), and Text2LIVE (Bar-Tal et al., 2022). To run SDEdit (Meng et al., 2022) on stable diffusion, we utilize the code provided in the Stable Diffusion official repo. The publicly accessible repository for VQGAN-CLIP (Crowson et al., 2022) is also applied in our experiments.

**Algorithm and Pseudo-code.** The algorithm of condition-flexible diffusion model (see §3.1) is demonstrated in Algorithm 1. The pseudo-code of instance normalization guidance in condition-flexible diffusion model is given in Pseudo-code 1. Modules for in-context visual programming are provided in in Pseudo-code 2. Our work is implemented in Pytorch (Paszke et al., 2019). Experiments are conducted on one NVIDIA TESLA A100-80GB SXM GPU.

**Reproducibility.** Our demo page is released at here. To further guarantee reproducibility, our full implementation and code are publicly released.

---

**Algorithm 1:** Condition-flexible diffusion model inversion

---

**Input:** A source prompt embedding $P$, translation textual embedding $\mathcal{P}$ and input image $\mathcal{I}$.
**Output:** Noise vector $z_T^*$.

---

Using instance normalization;
Compute the intermediate results $z_T^*, \ldots, z_0^*$ using diffusion model inversion over $\mathcal{I}$;
Using instance normalization;
Initialize $\bar{z}_T \leftarrow z_T^*$, $\varnothing_T \leftarrow \psi(\text{""})$;
**for** $t = T, T-1, \ldots, 1$ **do**
    **for** $j = 0, \ldots, N-1$ **do**
        $\varnothing_t \leftarrow \varnothing_t - \eta \nabla_\varnothing \|z_{t-1}^* - z_{t-1}(\bar{z}_t, \varnothing_t, P)\|^2$;
    **end**
    Set $\bar{z}_{t-1} \leftarrow z_{t-1}(\bar{z}_t, \varnothing_t, P)$, $\varnothing_{t-1} \leftarrow \varnothing_t$;
    Update $z_{t-1}^* \leftarrow \tilde{\epsilon}_\theta(\bar{z}_t, t, \mathcal{P}, \varnothing_t)$
**end**
**Return** $z_T^*$

---

**Pseudo-code 1:** Pseudo-code of instance normalization used in condition-flexible diffusion model in a PyTorch-like style.

```python
# cond_noise: conditional noise prediction embeddings
# uncond_noise: unconditional noise prediction embeddings

def calc_mean_std(input, eps=1e-8):
    #== Calculation of the mean and standard deviation of the input embedding
        ==#
    size = input.size()
    N, C = size[:2]
    input_var = input.view(N, C, -1).var(dim=2) + eps
    input_std = feat_var.sqrt().view(N, C, 1, 1)
    input_mean = input.view(N, C, -1).mean(dim=2).view(N, C, 1, 1)
    return input_mean, input_std

def instance_normalization(cond_noise, uncond_noise):
    #== Insntance normalization of shifting the distribution to conditional
        noise prediction ==#
    size = cond_noise.size()
    uncond_mean, uncond_std = calc_mean_std(uncond_noise)
    cond_mean, cond_std = calc_mean_std(cond_noise)

    normalized_noise = (uncond_noise - cond_mean.expand(size))/cond_std.expand(
        size)
    return conv(normalized_noise) * uncond_std.expand(size) + uncond_mean.expand
        (size)
```

## B MORE QUALITATIVE RESULTS FOR INSTANCE NORMALIZATION

In this section, we provided more qualitative results for instance normalization guidance (see §3.1) compared with different guidance scales $w$ in Fig 9. For fair comparison to other approaches, we do not utilize visual programming here (see §4.3). In addition, we report the overall qualitative results by integrating either traditional diffusion model or our proposed condition-flexible one within the framework of DVP. As shown in Fig. 11, our results consistently achieve robust performance with high fidelity when comparing with the fixed scale approach.

**Pseudo-code 2:** Pseudo-code of in-context visual programming in a PyTorch-like style.

```python
class VisualProgramming():
    def __init__(self):
    # load pre-trained models and transfer them to the GPU

    def html(self, inputs: List):
    # return an string that visually represents the input and output steps.

    def parse(self, plan: str):
    # analyze the step, extracting a list of input values and variables along
        with the name of the output variable

    def crop(self, image, plan: str):
    # return the image by cropping at the given coordinates and stores the
        coordinates as attributes

    def find(self, image, plan: str):
    # return a list of objects that match the object_name if any such objects
        are discovered

    def move(self, iamge, objects: Dict, plan: str):
    # return an image with objects that have been moved to new positions

    def execute(self, plan: str, state):
        inputs, input_var_names, output_var_name = self.parse(plan)

        # retrieve the values of input variables from the state
        for var_name in input_var_names:
            inputs.append(state[var_name])

        # execute the code and calculates the output utilizing the loaded model
        output = some_computation(inputs)

        # update the state
        state[output_var_name] = output

        # visual representation summarizing the computation of the step.
        step_html = self.html(inputs,output)

        return output, step_html
```

## C    MORE QUALITATIVE RESULTS FOR IN-CONTEXT REASONING

The DVP framework utilizes a series of visual programming operations to enable image translation, thereby enhancing its capacity for context-aware reasoning for arbitrary content manipulation. Fig. 10 provides more results and showcases our advantages over strong baselines, *i.e.*, Prompt2Prompt (Mokady et al., 2023) and VISPROG (Gupta & Kembhavi, 2023).

## D    VIDEO TRANSLATION

We further evaluate the efficacy of DVP on video translation in Fig. 12. Empirically, we are proficient in translating video content without the necessity for explicit temporal modeling. However, we do acknowledge certain issues during translation: a) the occurrence of artifacts, which depict non-existent elements within the original image (see □); and b) instances of missing objects (see □), often resulting from the failure to accurately identify the object, possibly due to the object possessing only a limited number of informative pixels or assuming an unconventional pose. More failure cases are discussed in §L.

## E    GENERAL-CONTEXT DATASET

We present a general-context dataset consisting of 500 image-text pairs across 20 different classes representing common objects in their typical contexts. Each class contains 25 unique samples. The objects span a range of common everyday items (see Fig. 13 top) such as foods, furniture, appliances, clothing, animals, plants, vehicles, *etc*. The images consist of high-quality photographs capturing the objects in natural environments and generated images with advanced digital rendering

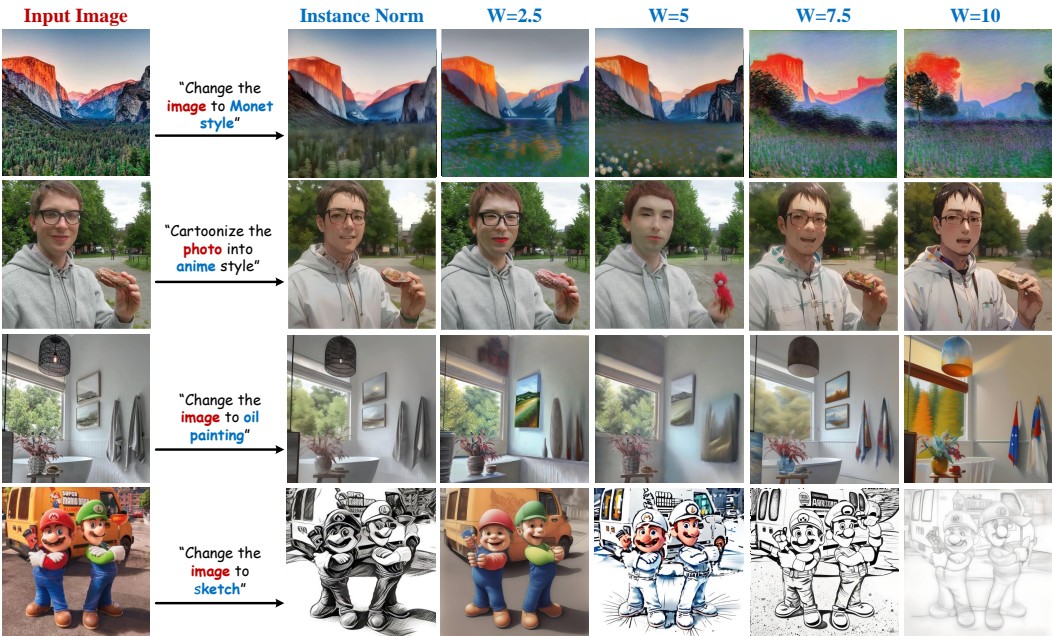

Figure 9: **More qualitative results** for instance-normalization.

techniques. The textual descriptions detail the location and spatial connections between primary objects and other context-rich elements in the scene produced by the GPT-driven prompter (see §3.2). Each final image-text pair (see Fig. 13 bottom) incorporates a source image with its descriptions, comprehensive instructions, and a translated image generated by DVP, which also includes cues about the position of objects in relation to other elements in the setting (*e.g.*, "Two foxes are standing to the **right** of a wolf that is perched on a rock in front of a forest"). The fine-grained details in both images and instructions aim to establish a comprehensive and demanding benchmark for future studies capable of precisely anchoring position relations within intricate real-world scenarios.

## F    RUNTIME ANALYSIS

We further do runtime analysis, shown in Table 3. As seen, though SDEdit achieves a superior performance in editing speed by excluding a diffusion inversion process, coming with a price that is less effective in retaining the intricate details from the original image. In contrast, our proposed method achieves satisfying image translation results with *limited* computational overhead (*e.g.*, $\sim 30\%$) when compared to other famous techniques, *i.e.*, Text2Live (Bar-Tal et al., 2022), VQGAN-CLIP (Crowson et al., 2022), and VISPROG (Gupta & Kembhavi, 2023). Specifically, Text2live (Bar-Tal et al., 2022) generates an edit layer of color and opacity, which is then composited over the original input image, rather than directly producing an edited output; VQGAN-CLIP (Crowson et al., 2022) employs a multimodal encoder to assess the similarity between a text-image pair and iteratively updates the candidate generation until it closely resembles the target text; VISPROG (Gupta & Kembhavi, 2023) introduces visual programming to compositional visual tasks relying solely on a standard stable diffusion model (Rombach et al., 2022). Compared to these methods, DVP facilitates various editing operations (see §3.2) during the translation process, adding to its practicality in applications where versatility in editing is crucial (see §4.2). We thus argue that the overhead of our proposed DVP is acceptable, considering its satisfying performance.

Table 3: **Runtime analysis.** We measure the total editing time for different methods.

| Method | Editing |
|---|---|
| SDEdit | $\sim 10$s |
| VQGAN-CLIP | $\sim 1.5$m |
| Text2live | $\sim 10$m |
| VISPROG | $\sim 1.5$m |
| Ours | $\sim 2$ m |

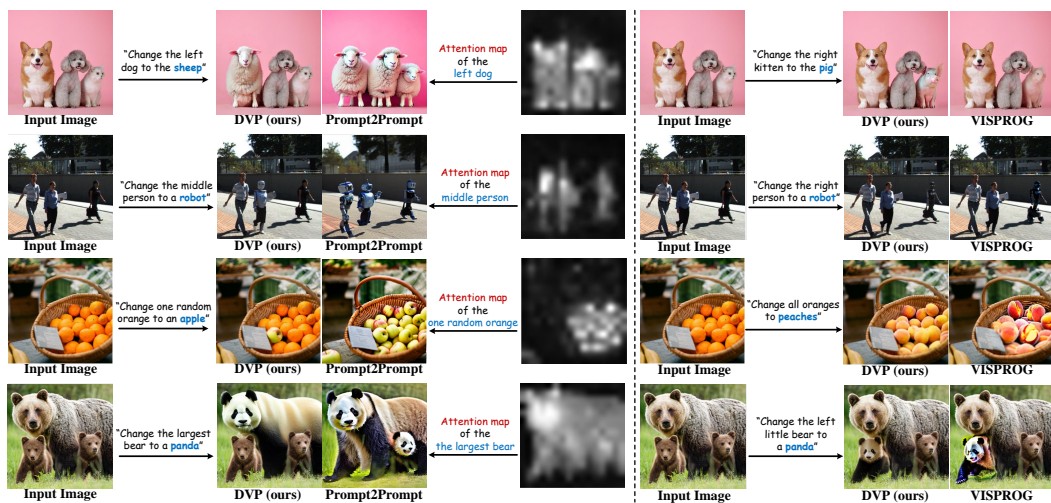

Figure 10: **More qualitative results** for in-context reasoning.

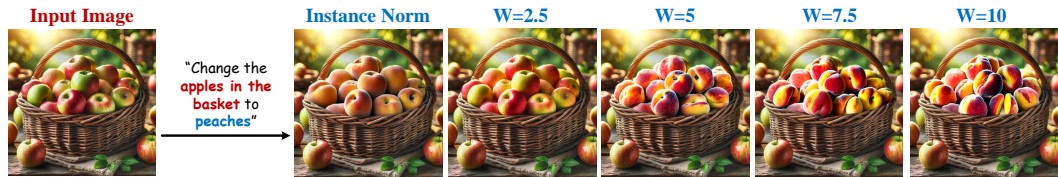

Figure 11: **Qualitative results** by integrating either traditional diffusion model or our proposed condition-flexible one (see §3.1).

## G  MORE QUALITATIVE RESULTS

We present more qualitative results in Fig. 14. As seen, our DVP is capable of achieving appealing performance in various challenging scenarios. For example, in the first row of Fig. 14, DVP transfers the husky into a sheep with high fidelity, and preserves all details from the intricate outdoor background.

## H  MODULE REPLACEMENT ANALYSIS

We also introduce a segmentation model replacement module, substituting the model (Cheng et al., 2022) with Segment Anything (Kirillov et al., 2023). As demonstrated in Fig. 15, Segment Anything exhibits superior segmentation capacity (correctly identifying a zebra with partial occlusion), resulting in improved translation results. Our modulating design enhances overall performance by replacing the segmentation component, particularly in scenarios that demand accurate per-pixel understanding. This characteristic ensures the flexibility of our approach, enabling effective management of a diverse range of scenarios (Wang et al., 2022; Qin et al., 2023; Liu et al., 2021; Han et al., 2023; Yan et al., 2023; Han et al., 2024).

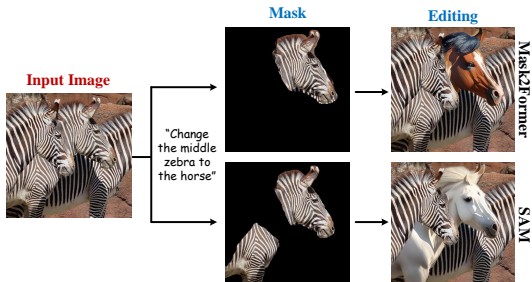

Figure 15: Module replacement analysis. We replace Mask2former (Cheng et al., 2022) to Segment Anything (Kirillov et al., 2023) as our *RoI Segmenter* in DVP.

## I    COMPARISON WITH INSTRUCT-PIX2PIX

In Fig. 16, we present a detailed qualitative comparison between our method and Instruct-Pix2Pix (Brooks et al., 2023). As seen, our approach yields better qualitative results. For example, when changing the kid to branches, DVP presents a more faithful translated result than Instruct-Pix2Pix. We also need to point out that Instruct-Pix2Pix (Brooks et al., 2023) relies on an instructive tuning-based diffusion model, which necessitates a rigorous and extensive training schedule. This stands in stark contrast to our method. Our paradigm is notably training-free, eschewing the need for such intensive data processing and model adjustment.

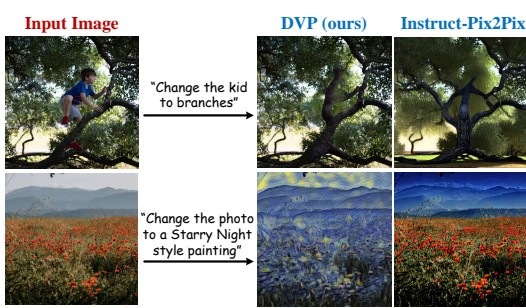

Figure 16: Qualitative comparison with Instruct-Pix2Pix.

This fundamental difference in approach not only sets our method apart but also underscores its efficiency where extensive training schedules are either impractical or undesirable.

## J    AMBIGUOUS INSTRUCTION ANALYSIS

To evaluate the robustness of DVP, we conducted an ablation study to determine the quality of the translated image under ambiguous instructions. The results in Fig. 17 show that despite the vagueness of the instructions, our approach is still capable of translating images with reasonably accurate results (*e.g.*, when having "apples" as the only instruction, DVP is able to translate both ice-cream and burger to the apples). The validity of translated results can be primarily attributed to the advanced capabilities of GPT planning (see §3.2). GPT's understanding of nuanced instructions and its ability to generate coherent plans from incomplete or unclear data played a pivotal role. Our

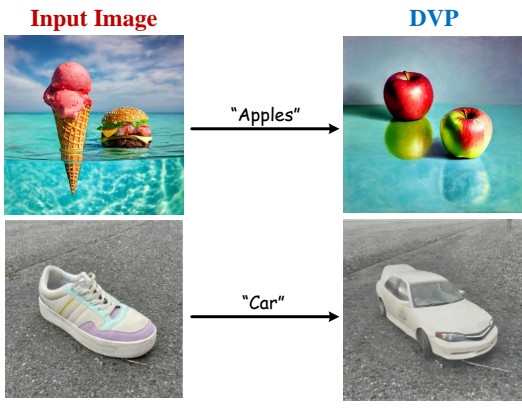

Figure 17: Ambiguous instruction analysis.

approach thus not only excels in straightforward scenarios but also possesses a commendable level of adaptability when faced with ambiguous or less-defined instructions. Such a feature is valuable in real-world applications where clear and precise instructions may not always be available.

## K    COMPLEX INSTRUCTION ANALYSIS.

DVP harnesses the strong capabilities of the GPT as its core planning mechanism. In Fig. 18, we demonstrate how DVP applies complex textual augmentations to translate the image effectively. In this example, the instruction is given as 'Swap the dog on the far right to the most left, and then swap the dog in the middle to the right" necessitating the planner to interpret and process the intricate/multiple textual

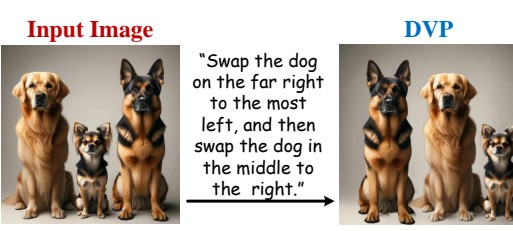

Figure 18: Complex textual arguments.

descriptions "to the most left" and "on the far right." DVP shows a proficient capability in comprehending the underlying textual descriptions and subsequently creating satisfying results. This illustration exemplifies how, by leveraging the advanced capabilities of GPT, our approach is also adept at managing and handling complex arguments.

## L   ERROR ANALYSIS

In Fig. 19, we present an overview of the most notable failure cases and derive conclusions concerning their distinct patterns that result in less-than-optimal outputs. Evidently, our DVP encounters difficulties in differentiating and translating objects from backgrounds with challenging situations, such as poor photometric conditions (Fig. 19 bottom) and occluded objects (see Fig. 19 top). For instance, in the first row, poor lighting causes the translation to appear odd; in the second row, under unfavorable photometric conditions, both shape and shading are mistakenly identified as objects and utilized for subsequent translation; in the third row, the cat is obscured, making its translation into a full tiger challenging; in the fourth row, due to the crowded nature of the lychees, their count varies after translation. The translation on these challenging situations remains a common limitations observed across the majority of concurrent diffusion models. Additional research in this domain is thus imperative. A viable approach involves the integration of amodal segmentation models (Zhang et al., 2019; Zhu et al., 2017b) (*i.e.*. considering both the visible and occluded parts of the instance) to fully restore object masks prior to the reconstruction process. We highlight this as a future direction for addressing challenging scenarios.

## M   USER STUDY

In Table 1, we design user study including "Quality," "Fidelity" and "Diversity," respectively. Specifically, we employ user study on Likert scale (Likert, 1932), a well-established rating scale metric utilized for quantifying opinions, attitudes, or behavioral tendencies (Allen & Seaman, 2007; Bertram, 2007; Boone Jr & Boone,

Table 4: **User study** on controllability and user-friendliness.

| Method | Controllability | User-friendliness |
|---|---|---|
| VISPROG | 25.1% | 39.7% |
| DVP (ours) | 74.9% | 60.3% |

2012; Norman, 2010). This scale typically presents respondents with either a statement or a question. In our case, the questions are measuring the overall harmony of the translated image (*i.e.*, "Quality"), the preservation of the identity in the image (*i.e.*, "Fidelity"), and variations between the original and translated images (*i.e.*, "Diversity") rated by users from 1 (worst) to 5 (best), Respondents then select the choice that most closely aligns with their own perspective or sentiment regarding the given statement or question.

We further provide user study in Table 4, focusing on the usability of explainable controllability discussed in §4.3. The results demonstrate that our approach facilitates user-friendly error correction and permits the intuitive surveillance of intermediate outputs.

## N   DISCUSSION

**Novelty.** Our DVP contributes on three distinct technical levels. Specifically, we first re-consider the design of classifier-free guidance, and propose a *conditional-flexible* diffusion model which eschews the reliance on the sensitive manual control hyper-parameter $w$. Second, by decoupling the intricate concepts in feature spaces into simple symbols, we enable a *context-free* manipulation of contents via visual programming, which is both controllable and explainable. Third, our GPT-anchored programming framework serves as a demonstrable testament to the versatile *transferability* of Language Model Models, which can be extended seamlessly to a series of critical yet traditional tasks (*e.g.*, detection, segmentation, tracking tasks). We do believe these distinctive technical novelties yield invaluable insights within the community.

**Limitation and Future Work.** Despite DVP showcases superiority over state-of-the-art methods in both qualitative and quantitative aspects (see §4.2), it also comes with new challenges and unveils some intriguing questions. For example, our approach grapples with obscured objects, attributed mainly to the segment module's limitation in exclusively processing the visible segment of an object, neglecting the occluded portion. This limitation consequently influences the diffusion process to generate base merely on the segmented part. In addition, in scenarios of photometric conditions, DVP and its counterparts falter in accurately aligning with the translated object(s), echoing challenges akin to the aforementioned issue (see §L). We posit that a specialized, fine-grained dataset tailored for a specific purpose (*e.g.*, occluded items, photometric conditions) might be the future direction to bolster the diffusion model's proficiency. In our study,

we integrated instance normalization with the null-text optimization, which is specifically design for text-guided diffusion model. This integration, however, is not directly applicable to a wider range of diffusion-based image generation applications. Nonetheless, the substantial potential of instance normalization has not been fully explored or utilized yet. A promising direction to leverage the power of instance normalization lies in fine-tuning the text-to-image task, which is split into a two-step procedure: image generation followed by image fine-tuning. Specifically, we can first generate some base images with any text2image model. Subsequently, we can naturally employ our approach with instance normalization to fine-tune these images. For example in Fig. 20, we initially produce an image following comprehensive instructions and then proceed to fine-tune the images directly, eliminating the need for complete regeneration the image from scratch.

This pipeline enhances the quality of the generated images, tailoring them to achieve the satisfying results. Moreover, fine-tuning existing images presents a more time and computationally efficient alternative compared to the regeneration from scratch at each time. This effort also

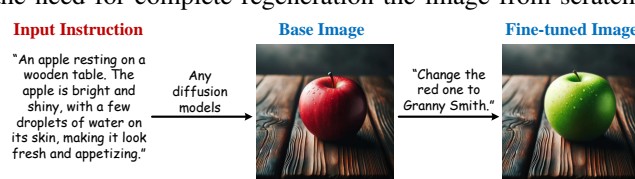

Figure 20: Expand instance normalization to broader image generation pipeline.

has the potential to handle image variations in lighting, contrast, and other characteristics.

**Societal Impact.** This work introduces DVP as a neuro-symbolic framework for image translation, showing robust image translation, strong in-context reasoning and straightforward controllability and explainability. On positive side, our framework reaches superior image translation performance qualitatively and quantitatively, and provide a user-centric design for the integration of future advanced modules. DVP holds significant merit, particularly in applications pertinent to safety-critical domains and industrial deployments. For potential negative social impact, our DVP struggles in handling obscured objects and photometric conditions, which are common limitations of almost all concurrent diffusion models. Hence its utility should be further examined.

**Ethics Concerns.** The inherent design of the DVP, characterized by its modular structure, implies that it does not intrinsically contribute knowledge or understanding to the process of image translation. This modular architecture indicates that DVP acts as a framework through which various processes are executed, rather than a knowledge-contributing entity in its own right. Therefore, the concerns related to ethics or bias within the DVP system are predominantly relevant to its individual modules, rather than the system as a whole. It is worth noting that the modular nature of DVP offers a significant advantage in addressing these biases and ethical concerns. Since each component operates as a discrete unit within the framework, it is feasible to identify and isolate the individual modules that exhibit concerns. Once identified, these biased modules can be replaced with neutral alternatives. This adaptability is crucial in maintaining the integrity and effectiveness of the DVP, ensuring that it remains a robust and fair tool in the realm of image translation.

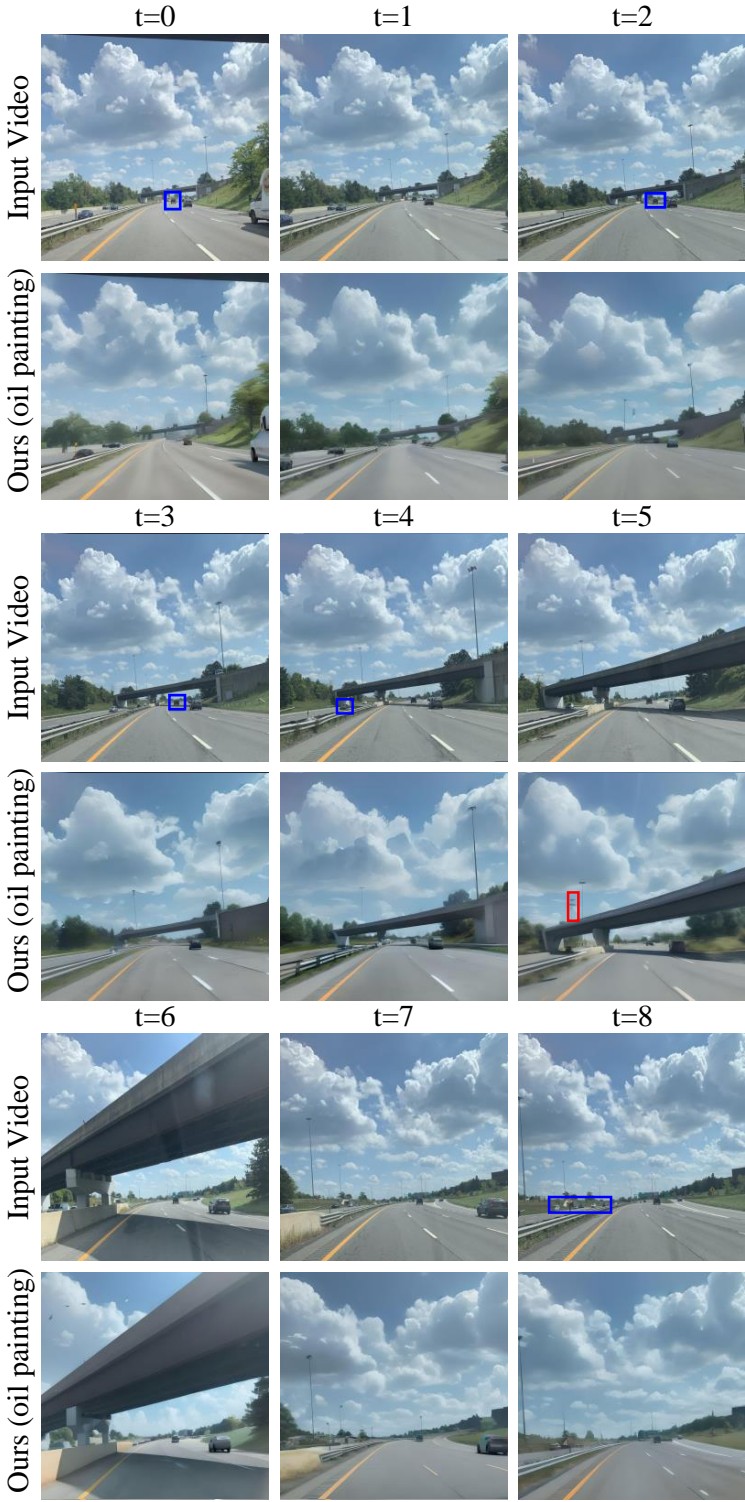

Figure 12: **Video translation results**. Devoid of any dedicated temporal module, our approach demonstrates commendable performance outcomes when applied to video data. □ and □ indicate artifacts and missing objects respectfully.

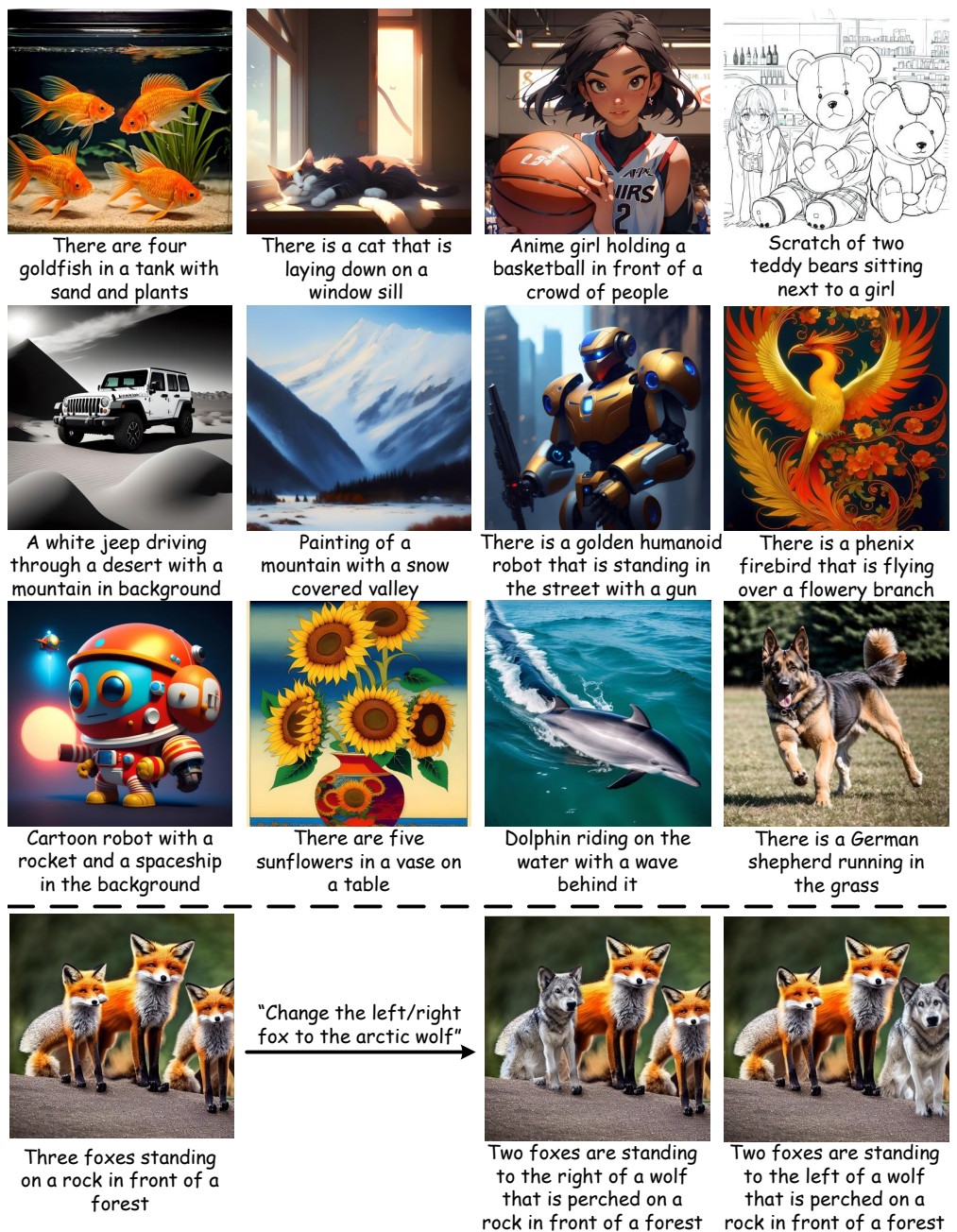

Figure 13: **General-context dataset** contains diverse image-text pairs (top three rows), and DVP presented images with targeted translation of the RoI (bottom two rows).

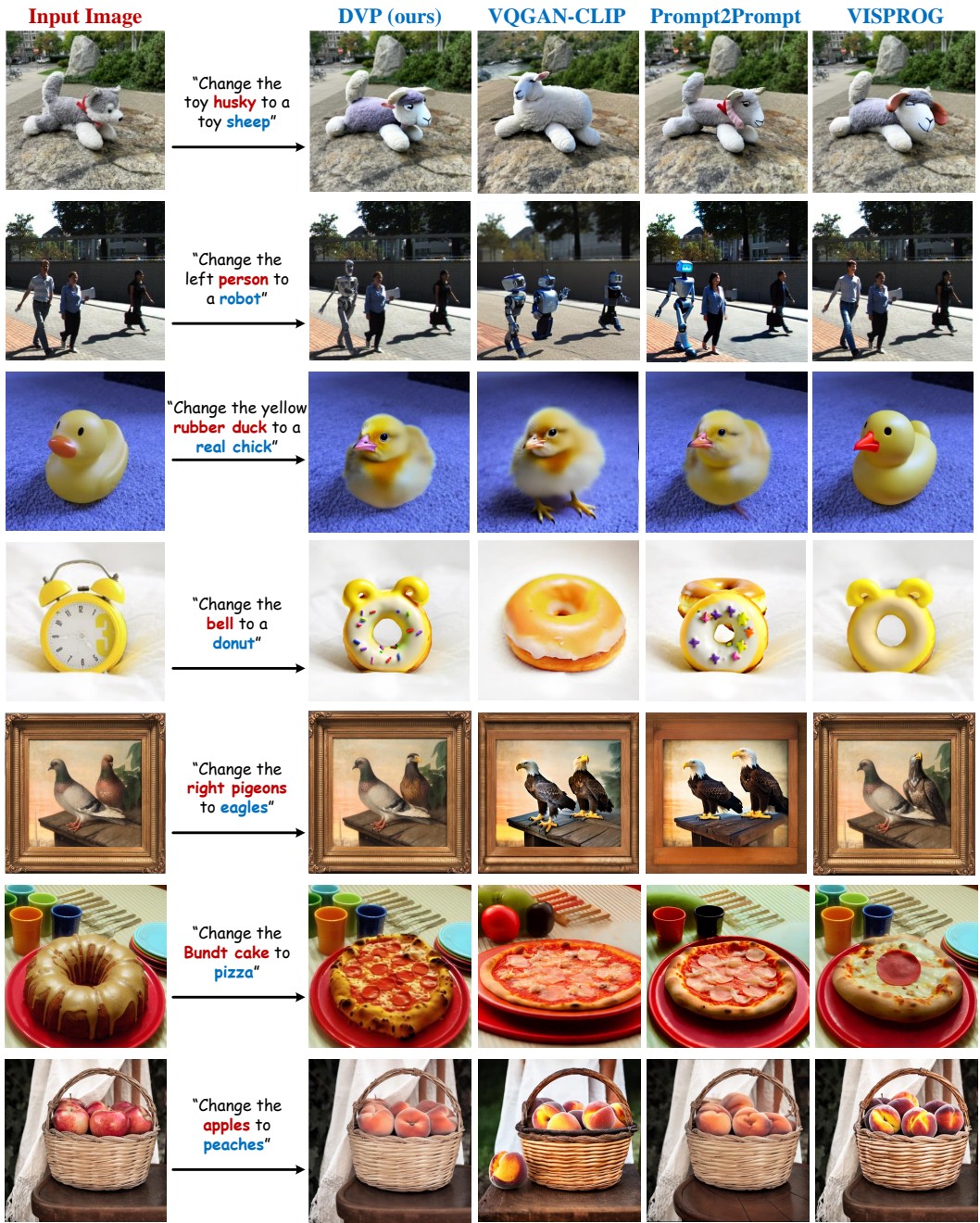

Figure 14: **More qualitative results of DVP.** As seen, DVP shows high-fidelity image translation with context-free manipulation.

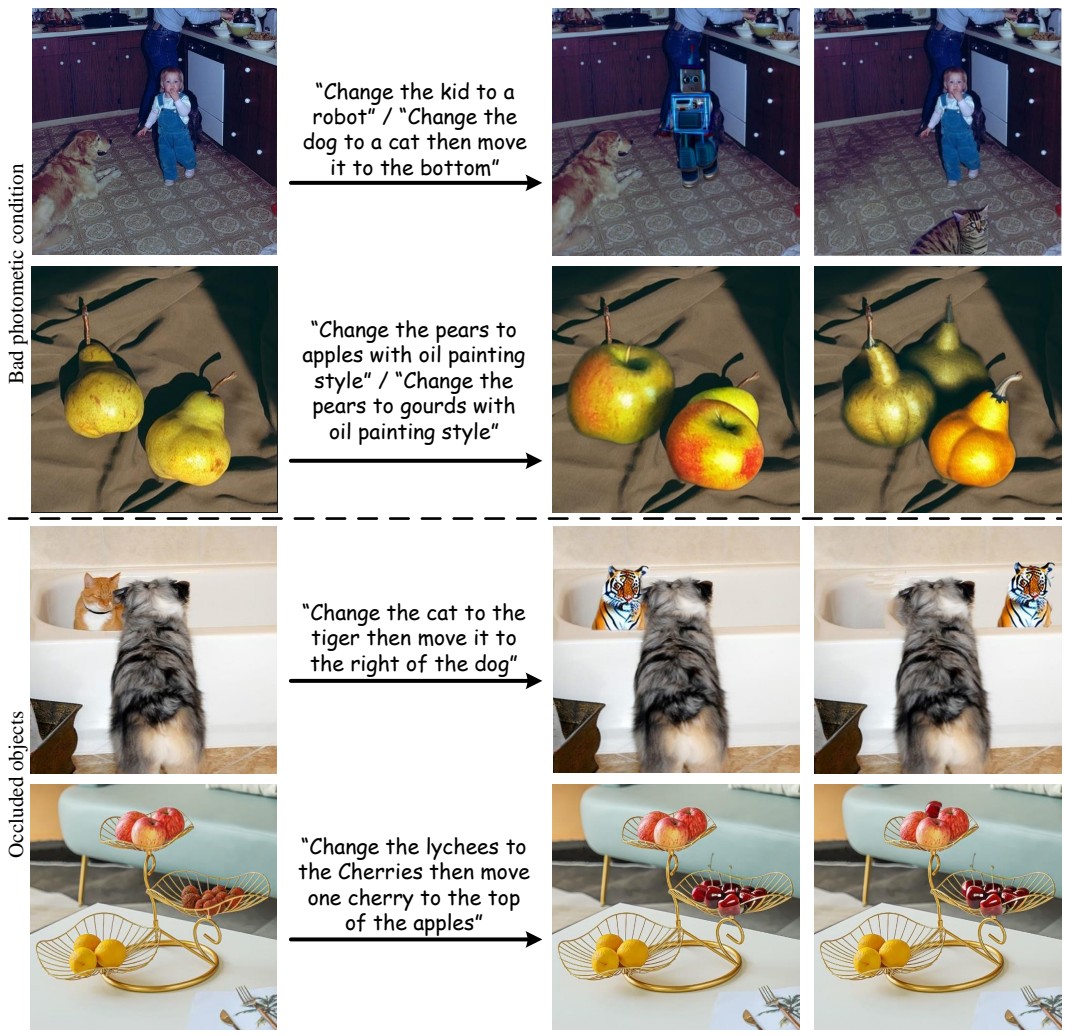

Figure 19: **Failure cases.** Current DVP might get unsatisfied results when the input image is in poor photometric conditions or contains occluded objects.

