# OpenReview forum: "Image Translation as Diffusion Visual Programmers"
_ICLR.cc/2024/Conference — ICLR 2024 poster_

### Official Review · Reviewer_EyuE · 2023-10-24

**Soundness:** 3 good
**Presentation:** 4 excellent
**Contribution:** 3 good
**Rating:** 6
**Confidence:** 4

**Summary:**

The paper presents the Diffusion Visual Programmer (DVP), a new neuro-symbolic framework for image translation that combines a diffusion model with the GPT architecture, coordinating a sequence of visual programs for image editing tasks like RoI identification, style transfer, and positioning. DVP stands out due to its ability to use instance normalization for condition-flexible translation, focusing on textual prompts for quality content creation, and its capability to convert complex concepts in feature spaces into simpler symbolic representations, ensuring localized editing that retains image coherence. Additionally, DVP provides clear symbolic representations at every stage, enhancing user control and understanding, marking progress in aligning artificial image processes with human cognitive intelligence.

**Strengths:**

The Diffusion Visual Programmer (DVP) method, rooted in a neuro-symbolic perspective, integrates a condition-flexible diffusion model with the GPT framework, guiding a series of pre-existing computer vision models to produce the desired results. DVP's strength lies in its ability to accurately position translated objects and allow for various context-free manipulations. The advantages of this work include:
1. Precision in Targeting: By prioritizing the identification of the Region of Interest (RoI), the system ensures accurate and relevant image translations.
2. High-fidelity Translations: After identifying the target region, the system maintains high-quality translations into the desired domain.
3. Spatial Awareness: DVP's capability to position translated objects based on spatial data enhances the translation's realism and context.
4. Enhanced User Control: The system's step-by-step approach allows users to have better control over the translation process, ensuring results align closely with their intentions.

**Weaknesses:**

1. Dependency on Modules: DVP's systematic design implies that if one module falters, the entire system could be compromised. The robustness of such an interconnected system becomes a pertinent question. If one cog in the machine is faulty, can the system still deliver optimal results, or will it be critically hampered?
2. Comparison with Other Techniques: Some results showcased by DVP can be achieved using established perception techniques like Segment Anything (SAM) or conditional control generation methods like ControlNet's point-to-point mode. A comparative discussion on the advantages of DVP over these techniques would provide clarity on DVP's unique value proposition.
3. Textual Argument Handling: Even though DVP marries a condition-flexible diffusion model with the GPT architecture, its prowess in handling complex textual arguments requiring intricate reasoning remains questionable. The paper does not offer substantial experimental evidence to vouch for the enhanced capabilities of integrating GPT, leaving room for skepticism.
4. Handling Complex Scenarios: DVP's recurrent challenge is its inability to adeptly manage intricate situations. As highlighted in the failure case study, the system falters in the face of occlusions, inadequate lighting conditions, and densely populated scenes. This limitation raises concerns about the system's applicability in real-world, diverse scenarios.

**Questions:**

See above

**Details Of Ethics Concerns:**

There is no discussion on the bias, e.g., instructions like "changing a man to a woman". What is the possible bias it may cause?

---

> ### Author Response · Authors · 2023-11-18
> **To Reviewer EyuE**
>
> Thank you for the valuable time and constructive feedback.
>
> #### **Q1 Dependency on Modules:**
>
> **A1:** Thank you for the great insight. The apprehension regarding the dependency of our proposed DVP on its constituent modules and the potential impact on the system's robustness is a critical aspect of our design considerations. In DVP, we integrate GPT ($i.e.,$ as a planner) and various built-in modules, where the interconnected nature is indeed a defining characteristic. DVP is designed with a modular architecture, where each component, while important, does not singularly dictate the system's overall functionality. The modular design allows for a quick debugging during the process. This is critical in maintaining the system's operational integrity even when individual components face challenges. We have discussed the failure and success cases in Fig. 7. However, if a module is **fully compromised**, the system's performance will be affected.
>
> #### **Q2 Comparison with Other Techniques**
>
> **A2:** Thank you for your insightful comments. We agree that established perception techniques like SAM generate promising results in various applications. These methods might not be directly comparable with DVP as they are not designed for image translation, and thus more suitable for replacing the current segmentation model ($i.e.,$ Mask2former [ref6]) for performance comparison. To understand how these models can further improve DVP, we conduct additional experiments by replacing our segmentation model with SAM. The preliminary results show that it indeed enhances the overall performance of DVP, indicating the flexibility of DVP for adopting established perception techniques. We’ve supplemented the additional experimental results in the Appendix Sec.H. Thank you for the great suggestion.
>
> #### **Q3 Prowess in handling complex textual arguments:**
>
> **A3:** Thank you for the great suggestion. DVP integrates the power from GPT as a planner, which is renowned for its ability to process and analyze complex textual data, forming the backbone of our approach. We’ve provided more experiments in Appendix Sec.K, focusing on complex instructions involving spatial relations.
>
> #### **Q4 Handle complex scenarios:**
>
> **A4:** This is a great observation. Regarding the handling of complex scenarios, such as those involving occlusions, the modular architecture of DVP is designed to be both scalable and adaptable. The current limitations observed in the failure case study are indeed points of concern, but they also serve as catalysts for ongoing improvements. Due to the modular nature, each module within DVP can be continuously developed and updated. This advantage allows us to progressively enhance the performance of the whole system, even in challenging conditions. Also, we need to emphasize that poor translation results in complex scenarios is a common problem in the concurrent community. As stated in Appendix Sec.L and N, we highlight it as a future direction of our DVP such as reconstructing the occluded objects before editing from additional amodal segmentation models for a better translation result.
>
> #### **Q5 Bias concerns:**
>
> **A5:**  Thank you for your concern. The nature of DVP as a modular system demonstrates that it does not inherently contribute knowledge during the image translation process. Consequently, concerns about bias are more pertinent to the individual modules we introduced in DVP and components it utilizes. For example, the diffusion model is found to exhibit biases [ref7-10] such as gender stereotypes or imbalances due to its training data. This problem can be solved by introducing/replacing a more neutral alternative [ref11-12]. We recognize the importance of this discussion and incorporate it into Appendix Sec.N. We also expect the computer vision society to **seriously** consider this concern.
>
> [ref1] AnyDoor: Zero-shot Object-level Image Customization. ArXiv
>
> [ref2] Inst-Inpaint: Instructing to Remove Objects with Diffusion Models. ArXiv
>
> [ref3] Visual Programming: Compositional visual reasoning without training. CVPR 2023
>
> [ref4] Segment anything. ArXiv
>
> [ref5] Adding Conditional Control to Text-to-Image Diffusion Models. ICCV 2023
>
> [ref6] Masked-attention Mask Transformer for Universal Image Segmentation. CVPR 2022
>
> [ref7] Stable bias: Analyzing societal representations in diffusion models. ArXiv
>
> [ref8] Analyzing bias in diffusion-based face generation models. ArXiv
>
> [ref9] Do ImageNet Classifiers Generalize to ImageNet? ICML 2019
>
> [ref10] Unbiased look at dataset bias. CVPR 2011
>
> [ref11] Fine-tune language models to approximate unbiased in-context learning. ArXiv
>
> [ref12] A Conservative Approach for Unbiased Learning on Unknown Biases. CVPR 2022
>
> We appreciate your thoughtful comments. The above discussions are incorporated in our revised paper (in orange). We hope our response addresses your concerns. Please let us know if there are any additional questions, and we are happy to discuss further.

---

### Official Review · Reviewer_DvDX · 2023-10-27

**Soundness:** 3 good
**Presentation:** 3 good
**Contribution:** 2 fair
**Rating:** 6
**Confidence:** 3

**Summary:**

This paper introduces a new method, named Diffusion Visual Programmer (DVP), for image translation. DVP is a neuro-symbolic framework that incorporates condition-flexible translation into an LLM planner. Specifically, the authors propose instance normalization to avoid manual guidance to achieve higher-quality content editing. The authors also prompt LLM to generate a sequence of programs with operations, which can be executed to plan the procedure, segment the object of interest, give the caption for the image, inpaint the masked image, and manipulate the sizes and positions of objects. DVP makes the process of image translation controllable and interpretable. The qualitative and quantitative results demonstrate a good performance of DVP.

**Strengths:**

- The paper integrates the condition-flexible diffusion model into a visual programming framework to improve image translation.
- The paper proposes an efficient instance normalization to improve the quality of image editing.
- GPT planner provides controllability and explainability for the image translation process.
- The generated results are good with quantitative and qualitative evaluation.
- The paper is well-written and easy to read.

**Weaknesses:**

- It seems the conditional-flexible diffusion model is the prompt-to-prompt method with proposed instance normalization. However, the authors only compare the results of using instance normalization and different guidance scales. It is also important to ablate the instance normalization in DVP to demonstrate instance normalization significantly outperforms a fixed scale when incorporating in-context visual programming.
- The proposed method is an aggregation of different models planned by an LLM. However, how the components (e.g., mask2former, Repaint, BLIP, position manipulator) are aggregated is not clearly clarified. The authors may explain various translation procedures by providing more step-wise illustrations like Fig. 7. Besides, the authors may provide the prompt they use for the GPT planner.
- Some important details are missing. For example, how the conv layer in instance normalization is parameterized? Is there any other difference between the conditional-flexible diffusion model and the prompt-to-prompt model, except for instance normalization? How to decide the order of operation sequences, automatically by GPT or manually by humans?

**Questions:**

- The authors use a conv layer in instance normalization (Eq. 5). But there is no evidence of how the conv layer is trained or parameterized. Can authors give more details?
- In Figure 7, We can see different plans derive different translated results. How do authors obtain different orders of operation sequences? How do authors decide which plan is the best for the final result?
- In Fig. 8, are reconstructed results with automatic labels and human labels generated by null-text inversion? To my knowledge, null-text inversion produces quite similar reconstructed images to the original ones regardless of the input prompt. Can authors explain why using GPT-generated prompts derive better reconstructed results than human annotations?

---

> ### Author Response · Authors · 2023-11-18
> **To Reviewer DvDX**
>
> We thank reviewer DvDX for the valuable time and constructive feedback.
>
> #### **Q1 Additional ablation for instance normalization**
>
> **A1:** Thank you for the great suggestion. We have conducted an additional ablation by comparing instance normalization in DVP with a fixed scale. Our results show that instance normalization significantly outperforms a fixed scale when incorporating in-context visual programming, indicating the effectiveness of instance normalization. We have also included the additional results and discussions in Appendix Sec.B Figure 11.
>
>
> #### **Q2 More details on component aggregation and prompt for GPT planner:**
>
> **A2:** Thank you for the suggestion. We leverage the capabilities of GPT to create a sequence of operations. This planning process yields a sequence of distinct interpreters, such as SegmentInterpreter and EDITInterpreter. The method of translating is identical to that shown in Figure 7. For the prompts specified in the GPlan, we instruct GPT-4 to generate code that adheres to the same format employed for each individual pre-defined component. We’ve provided more clarification on how the components are aggregated in the revision.
>
>
> #### **Q3.1 More details on Conv layer in instance normalization:**
>
> **A3.1:** As outlined in our implementation, we employ the null-text optimization method [ref1] to tune the conv layer. This approach involves optimizing the convolutional layers and the unconditional embedding, initialized with a null-text embedding. This technique, as we have employed, ensures a balance between high-quality image reconstruction and the provision of intuitive editing capabilities. We’ve added more details in the revision.
>
>
> #### **Q3.2 Decision-making for the order of operation sequences:**
>
> **A3.2:** Thanks for the insightful question. The process of obtaining these various sequences of operations is primarily facilitated by GPT-4, which acts as an automated planner. This automated generation often results in a diverse array of operation sequences, each potentially leading to distinct outcomes. Additionally, we can also force GPT-4 to follow a pre-defined order of operations by providing proper instruction.
>
> In our current framework, we do not definitively conclude which plan is superior. Instead, GPT-4's role as an automated planner is to provide a range of reasonable and logically structured operation sequences. However, identifying the ‘’best’’ plan for the results remains an open question for our future exploration.
>
>
> #### **Q4 Fig. 8:**
>
> **A4:**  Sorry for the confusion. The reconstruction results shown in this figure use only 40% of the optimization steps. We aim to demonstrate the advantage of our method (fine-grained accurate annotation) over the baseline using a much shorter schedule. We acknowledge the necessity to revise the image description in our paper to clearly delineate this distinction (added in revision paper Sec. 4.3). Regarding the full null-text inversion process, it generally yields reconstructed images that closely match the originals, regardless of the input prompt, even when the prompts are entirely incorrect.
>
> [ref1] Null-text inversion for editing real images using guided diffusion models. CVPR 2023
>
> We appreciate your thoughtful comments. The above discussions are incorporated in our revised paper (in green). We hope our response addresses your concerns. Please let us know if there are any additional questions, and we are happy to discuss further.

---

> > ### Comment · Reviewer_DvDX · 2023-11-22
> >
> > Thank the authors for providing their rebuttal. I have no further questions.

---

### Official Review · Reviewer_TXue · 2023-10-28

**Soundness:** 3 good
**Presentation:** 3 good
**Contribution:** 3 good
**Rating:** 6
**Confidence:** 4

**Summary:**

In this paper, the authors introduce the Diffusion Visual Programmer (DVP), a neuro-symbolic image translation framework that seamlessly combines a diffusion model with the GPT architecture. DVP enables transparent and controllable image translation processes, covering tasks like RoI identification, style transfer, and position manipulation. Extensive experiments showcase DVP's remarkable performance, surpassing existing methods. The success of DVP can be attributed to its condition-flexible translation, in-context reasoning, and systemic controllability. This research represents a significant advancement in harmonizing image translation with cognitive intelligence, with promising applications in various domains.

**Strengths:**

1. **Innovative Framework**: DVP introduces a novel neuro-symbolic image translation framework, seamlessly combining a diffusion model with the GPT architecture for various image processing tasks.

2. **Remarkable Performance**: DVP outperforms existing methods, demonstrating high-quality image translation through extensive experiments.

3. **Condition-Flexible Translation**: DVP achieves condition-flexible translation via instance normalization, enhancing content generation while reducing manual guidance sensitivity.

**Weaknesses:**

1. **Novelty**: The paper should address concerns about its novelty, as it resembles existing visual programming pipelines with the primary distinction being the choice of image editing tools. Clarification on how DVP significantly sets itself apart from previous work is crucial.

2. **Lack of Comparative Analysis**: The paper's strength in performance could be further substantiated by a more thorough comparative analysis. A comparison with alternative approaches like instructive tuning-based diffusion models, such as instruct2pix, would provide a clearer perspective on its relative merits.

3. **Limited Experimentation on Instance Normalization**: The paper introduces instance normalization as a valuable element of the DVP framework, which can have broader applications. However, there is a need for more extensive experimentation to validate the advantages of instance normalization in the context of general diffusion-based image generation.

**Questions:**

While the paper showcases DVP's impressive performance, there is a need for further exploration of its adaptability to a range of image translation tasks. These tasks might include object replacement, color/texture/material editing, and style transfer. Currently, the paper's experiments encompass a holistic approach without distinct analysis of how DVP caters to varying task requirements.

Additionally, the paper's reliance on a relatively small self-collected dataset comprising 100 image-text pairs may present limitations in terms of the breadth and diversity of data for evaluation.

---

> ### Author Response · Authors · 2023-11-18
> **To Reviewer TXue**
>
> We thank reviewer TXue for the valuable time and constructive feedback.
>
> #### **Q1 Novelty of DVP:**
>
> **A1:** Thank you for the question. We try to provide some clarification here.
>
> **Limitation of existing work**: Current image translation approaches ($i.e.,$ AnyDoor and Inst-Inpaint) remain **opaque** [ref1] or need **additional training** [ref2]. Specifically, for AnyDoor, it remains untransparent with an additional ID extractor, and detailed maps for hierarchical resolutions, preventing it from intuitively manual modifications. For Inst-Inpaint, it required extra training, which is not amenable to the training-free paradigm. Visual Programming, on the other hand, is a neuro-symbolic approach available for image translation. However, the strong editing abilities for **compositional generalization have been overlooked**.
>
> **DVP contribution**: Our DVP contributes on three distinct technical levels. First, we re-consider the design of classifier-free guidance, and propose a **conditional-flexible** diffusion model which eschews the reliance on sensitive manual control hyper-parameters. Second, by decoupling the intricate concepts in feature spaces into simple symbols, we enable a **context-free** manipulation of contents via visual programming, which is both controllable and explainable. Third, our GPT-anchored programming framework serves as a demonstrable testament to the versatile **transferability** of Language Model Models, which can be extended seamlessly to a series of critical yet traditional tasks ($e.g.,$ detection, segmentation, tracking tasks). We do believe these distinctive technical novelties yield invaluable insights within the community.
>
> We add discussion in Appendix Sec.N in our revision paper to highlight the novelty of our approach. Thank you for the suggestion!
>
> #### **Q2 Lack of Comparative Analysis:**
>
> **A2:** Thank you for the great suggestion. We agree that additional comparison with instructive tuning-based diffusion models would provide a clearer understanding of the advantage of our model. We conduct a thorough comparative study with instruct2pix [ref3] as suggested. As shown in Fig. 16, our approach consistently yields high-quality performance due to its robust capacity for in-context reasoning. We have included these qualitative findings in Appendix Sec.I.
>
> #### **Q3 Instance normalization can have broader applications.**
>
> **A3:** We appreciate the insightful suggestion. In our work, instance normalization is specifically coupled with the null-text optimization technique, which is tailored to suit the unique requirements of our approach and is not yet generalized for broader diffusion-based image generation applications. However, we do recognize that instance normalization harbors a considerable potential that remains largely untapped. Therefore, we conduct a preliminary study for the text-to-image task. Specifically, we initially generate an image following comprehensive instructions and then proceed with the fine-tuning process, eliminating the need for complete regeneration of the image from scratch. These preliminary results together with the discussions are supplemented in Appendix Sec.N and Fig. 20.
>
> #### **Q4 Adaptability to a range of image translation tasks:**
>
> **A4:** We showcase several examples of object replacement in specific locations to highlight the robust in-context reasoning capability of the proposed DVP, addressing limitations observed in previous methods. It's important to note that DVP is a general framework for image translation, applicable to various task variants. To further demonstrate the adaptability of DVP, we provide additional results and discussions on style transfer (see Fig. 9 in the Appendix) and editing (refer to Figures 14, 16, 17, and 18 in the Appendix) as suggested.
>
> #### **Q5 The dataset is self-collected:**
>
> **A5:** Due to the scarcity of publicly accessible datasets, we have introduced a new dataset featuring detailed textual descriptions. For quantitative evaluation, we adhere to established methods [ref1, ref4-5] and execute these on our newly developed dataset, detailed in Appendix Sec.E. To ensure a level playing field, we apply all methods uniformly across our dataset. Additionally, we plan to expand our data collection to encompass a broader diversity, and release our dataset.
>
>
> [ref1] AnyDoor: Zero-shot Object-level Image Customization. ArXiv
>
> [ref2] Inst-Inpaint: Instructing to Remove Objects with Diffusion Models. ArXiv
>
> [ref3] InstructPix2Pix: Learning to Follow Image Editing Instructions. CVPR 2023
>
> [ref4] Prompt-to-Prompt Image Editing with Cross Attention Control. ICLR 2023
>
> [ref5] Text2LIVE: Text-Driven Layered Image and Video Editing. ECCV 2022
>
> We appreciate your thoughtful comments. The above discussions are incorporated in our revised paper (in blue). We hope our response addresses your concerns. Please let us know if there are any additional questions, and we are happy to discuss further.

---

> > ### Comment · Reviewer_TXue · 2023-11-20
> > **Further question**
> >
> > Regarding Question 1, I would like to clarify my query. My original question pertained to how your approach differs from **visual programming**, not from other generative models. Since visual programming also supports image editing, could you specify what `additional tasks or capabilities your method offers that are not possible` with traditional visual programming?

---

> ### Author Response · Authors · 2023-11-20
> **Thank you for your additional question**
>
> Sorry for our misunderstanding, and thank you for the clarification. In comparison to traditional visual programming, our approach demonstrates several distinct capabilities:
>
> 1. **Arbitrary Positional Editing**: Our system empowers unrestricted editing, including the manipulation of position, thanks to our advanced **in-context reasoning** capability—a functionality that is not achievable in traditional visual programming. More precisely, we transform human instructions into a domain-specific language, incorporating a comprehensive set of fundamental logic (refer to Sec 3.2 and Figure 7), thereby facilitating versatile image editing.
>
> 2. **Automatic Image Translation**: Traditional visual programming relies on off-the-shelf modules that depend on manually crafted guidance scale parameters to oversee the translation process for each individual image, resulting in **condition-rigid learning**. In contrast, our DVP introduces a novel **condition-flexible** diffusion model that operates fully automatically, eliminating the need for human intervention (refer to Sec. 3.1). This model outperforms traditional methods by robustly translating images without the need for tunable parameters, representing a substantial advancement in both the quality and versatility of image processing.
>
> 3. **Generalizability**: The DVP framework we present is highly flexible and extensible, adept at addressing a diverse set of tasks that extend beyond the traditional scope. This encompasses, but is not restricted to, video editing (refer to Appendix D Figure 12), text-to-image generation (refer to Appendix N and Figure 20), etc. The adaptability of our framework is underscored by its **compatibility** with various off-the-shelf models. This adaptability is largely attributed to our GPT planner, which seamlessly integrates with these models, thereby broadening the potential applications of our system.
>
> Thanks again for your thoughtful question! We are happy to discuss more if you have any other questions.

---

> > ### Comment · Reviewer_TXue · 2023-11-20
> > **Thanks for the Reply**
> >
> > Thank you for your response. However, I must point out that points 1 and 3 do not seem accurate. For point 1, the visual programming paper already allows for position selection through a detection model, making the current extension seem straightforward.
> >
> > Regarding point 3, the tasks related to image editing and generation you mentioned are also covered in that paper. While the video aspect isn't, it appears to be a simple modification and doesn't seem to offer technical or conceptual novelty.
> >
> > Point 2 is valid, as your paper introduces automatic guidance scaling.
> >
> > Thank you for addressing these points. I will consider them further.

---

> ### Author Response · Authors · 2023-11-20
> **Thanks for the further discussion**
>
> We really appreciate your prompt response, and the great insights.
>
> Regarding point 1, we fully concur that visual programming can indeed handle position selection. However, the key distinction lies in DVP's unique capacity for in-context reasoning when confronted with intricate and arbitrary position manipulations, a feat where traditional VP may falter. For instance, consider the task of 'changing the rightmost dog to a cat and shifting it to the leftmost'. This demands the model to accurately comprehend the underlying concepts and interpret their positional information/relations. We should have articulated this capability as in-context reasoning to make it more accurate.
>
> Regarding point 3, yes, we agree that VP can be tailored for such tasks. Our emphasis was on underscoring the **flexibility and extensibility** inherent in DVP, which is more easily extendable to new tasks compared to VP, attributed to our GPT planner.
>
> Thank you very much for pointing these out. We will incorporate the above discussion into the revision to make it more clear.

---

> > ### Comment · Reviewer_TXue · 2023-11-22
> > **Maintain my Original Score**
> >
> > Thanks to the author for their detailed explanation. I will keep my original score at 6. Good luck.

---

> > > ### Author Response · Authors · 2023-11-22
> > > **Thank you**
> > >
> > > We are genuinely grateful for your thoughtful feedback. Your opinions have significantly elevated the quality and clarity of our paper. Should you have any last-minute questions, please do not hesitate to reach out to us.
> > >
> > > Best, \
> > > Authors

---

### Official Review · Reviewer_fWRk · 2023-11-01

**Soundness:** 3 good
**Presentation:** 3 good
**Contribution:** 3 good
**Rating:** 6
**Confidence:** 3

**Summary:**

In this paper, the authors propose a new framework called Diffusion Visual Programmer (DVP) for image translation tasks. It follows the diagram that first identifies the instructed target region and then translates it into the targeted domain. In particular, the authors use instance normalization for context-flexible translation, avoiding adjusting guidance manually. Besides, the authors decouple the concepts in feature space into simple symbols and then deal with the symbols with visual programming. Qualitative and qualitative results show that DVP outperforms other state-of-the-art methods, providing more reliable, controllable, and interpretable image translation.

**Strengths:**

- Overall, the paper is well-organized and easy to follow. The figures and tables are informative.

- The performance of the proposed method is promising. Figures 4, 6 clearly demonstrate the superiority of DVP.

- The ablation study and system analysis are clear and informative, making it easy to see the effectiveness of different parts, such as instance normalization, and prompter.

**Weaknesses:**

- The proposed method is a systematic approach for image translation tasks incorporating different components. A potential drawback is its inference speed. It would be beneficial if the authors could compare inference speed with other image translation tasks.
- The comparison with methods like SDEdit, Prompt2Prompt, and InstructPix2Pix is somehow unfair since they do not require an additional segmentation network.
- The quantitative evaluation is only the proposed dataset, which contains fine-grained edit instructions. The effectiveness of DVP could be further proved by evaluating simple or even ambiguous instructions.

**Questions:**

- I have questions about the learning process of the 1×1 conv layer in equation (5). How is it exactly trained? And is it sensitive to the training sample size?
- Will instance normalization also work in text-to-image tasks? It will be interesting to see if it could generate higher fidelity images with semantic meaning more aligned with the provided text prompts.

---

> ### Author Response · Authors · 2023-11-18
> **To Reviewer fWRk**
>
> We thank reviewer fWRk for the valuable time and constructive feedback.
>
> #### **Q1 Inference speed with regard to other methods:**
>
> **A1:** Thank you for the great suggestion. The inference speed should be highlighted in the image translation task. We thus compare our proposed DVP with the baselines in the Table below.
>
> | Methods  | Editing time |
> | :-: | :-: |
> | SDEdit|  ~ 10s |
> | VQGAN-CLIP | ~ 1.5m |
> | Text2live|  ~ 10m |
> | VISPROG|  ~ 1.5m |
> | **Ours**|  ~ 2m |
>
> Related experiments and discussions are added in Appendix Sec.F.
>
> #### **Q2 Comparison to other methods somehow unfair:**
>
> **A2:** Thank you for the question. We try to provide some clarification here. First, we agree that these text-guided diffusion models are in a compact framework without additional segmentation networks. The reason is that they include the cross-attention maps for spatial configuration and geometry during editing, making it intricate to decompose them with an extra RoI identification through a segmentation network. Second, to get a more fair comparison, we’ve compared with VISPROG [ref1], which includes Mask2Former [ref2] as the segmentation network. The results in Sec. 4 show superior performance of our method both quantitatively and qualitatively. Third, we realize that AnyDoor [ref3] is another method that uses a segmentation module. However, we haven’t compared it since the code is currently not available. We appreciate your suggestion and plan to conduct more comparisons with this method in the future.
>
> #### **Q3 Effectiveness on simple or ambiguous instructions:**
>
> **A3:** Thank you for the excellent suggestion. We conducted an additional ablation test on DVP under conditions of ambiguous instructions. Specifically, we limited the instructions for the GPT planner to a single word. The results are promising, showing that DVP is able to handle simple or ambiguous instructions. The details of these experiments are provided in the Appendix, Sec.J. We plan to conduct more experiments in this direction.
>
> #### **Q4 More details on training of 1×1 conv layer:**
>
> **A4:** As stated in Sec. 2, our approach does not necessitate a full training phase. Instead, we include a pivotal optimization process ($i.e.,$ null-text optimization [ref1]) for each image during inference. More specifically, for each input image, our approach optimizes the 1x1 convolutional layer and the unconditional embedding concurrently, which initially starts from a null-text embedding. Therefore, it is not sensitive to the training sample size. Sorry for the confusion. We’ve supplemented the above discussion in Appendix Sec.A to make it more clear.
>
> #### **Q5 Extension of instance normalization to text-to-image tasks:**
>
> **A5:** Thanks for the insightful question. The instance normalization is not directly applicable to text-to-image tasks, as it is integrated under null-text optimization [ref4], which requires pivotal optimization between the image and descriptive text before initiating the translation process. However, instance normalization can be effectively utilized in text-to-image tasks with additional fine-tuning of the produced images. We have elaborated on the prospective developments of instance normalization in Appendix Sec.N. We believe that instance normalization holds significant, yet largely unexplored potential, particularly in various applications.
>
> [ref1] Visual Programming: Compositional visual reasoning without training. CVPR 2023
>
> [ref2] Masked-attention Mask Transformer for Universal Image Segmentation. CVPR 2022
>
> [ref3] AnyDoor: Zero-shot Object-level Image Customization. ArXiv
>
> [ref4] Null-text inversion for editing real images using guided diffusion models. CVPR 2023
>
> We appreciate your thoughtful comments. The above discussions are incorporated in our revised paper (in red). We hope our response addresses your concerns. Please let us know if there are any additional questions, and we are happy to discuss further.

---

> > ### Comment · Reviewer_fWRk · 2023-11-21
> > **Thanks for your response.**
> >
> > After reading the authors' response, most of my concerns are solved, including the fairness of the comparison, inference speed, and the possibility of using instance normalization in other tasks. Thus, I am happy to raise my rating to accept.

---

> ### Author Response · Authors · 2023-11-21
> **Thank you for your response**
>
> Thank you for your prompt response. We are genuinely grateful for your thoughtful feedback. We are really appreciative of the discussions, as they clearly strengthen the completeness, and further illuminate the future direction of our work.
>
> Best,
>
> Authors

---

### Author Response · Authors · 2023-11-18
**To All Reviewers**

Dear Reviewers,

Sorry for the delayed response, as we are preparing the additional experiments. We sincerely thank all reviewers for your valuable suggestions and constructive feedback. We have revised our paper accordingly. The major changes are as follows:

1. We’ve supplemented runtime analysis and discussion in Appendix Sec.F, Table 3, as suggested by Reviewer fWRk.
2. We’ve conducted additional experiments with ambiguous instructions and added the result and discussion in Appendix Sec.J, Fig. 17, by Reviewer fWRk's suggestion.
3. We’ve provided details about the optimization process for our implementation in Appendix Sec.A, as suggested by Reviewer fWRk and DvDX.
4. We’ve provided a discussion with the potential extension and future work of our approach in Appendix Sec.N, Fig. 20, as suggested by Reviewer fWRk and TXue.
5. We’ve added additional comparisons with Instruct-Pix2Pix in Appendix Sec.I, Fig. 16, as suggested by Reviewer TXue.
6. We’ve included experiments with complex instructions of spatial relation in Appendix Sec.K, Fig. 18, as suggested by Reviewer EyuE.
7. We’ve provided additional discussion with ethical concerns in Appendix Sec.N, as pointed out by Reviewer EyuE.

Sincerely yours,

Authors.

---

### Author Response · Authors · 2023-11-21
**Seeking an open dialogue**

Dear Reviewers,

We sincerely appreciate the time and effort you've devoted to reviewing our work. We understand that your schedule may be quite busy. As the authors-reviewer discussion phase draws to a close, we kindly request your attention to our responses (special thanks to Reviewer TXue, who has initiated an insightful discussion). Our aim is to gain insights into whether our responses effectively address your concerns and to ascertain if there are any additional questions or points you would like to discuss.

We look forward to the opportunity for further discussion with you. Thank you for your thoughtful consideration.

Best regards, \
The Authors

---

### Author Response · Authors · 2023-11-23
**Summary about author-reviewer discussion**

Dear Area Chair and Reviewers,

We would like to express our sincere gratitude for your efforts in facilitating the discussion regarding our paper. As the discussion is coming to an end, we would like to provide a brief summary of the key points that have been discussed:
- We have addressed the concern on runtime analysis by reporting additional results in Appendix Sec.F, Table 3, as suggested by Reviewer fWRk. In response to the case that the instructions are ambiguous, we conducted the experiment and added discussion in Appendix Sec.J, Fig. 17. We further include the discussion on the optimization process and potential extension in Appendix Sec.A and Sec.N Fig 20, respectively. We are pleased to note that Reviewer fWRk has acknowledged that our response adequately addressed the concerns and subsequently increased the rating to reflect this.
- We have added additional comparisons with Instruct-Pix2Pix in Appendix Sec.I, Fig. 16, as suggested by Reviewer TXue. Furthermore, we discussed the novelties and potential extension of instance normalization. We thank Reviewer TXue for all the insightful discussion, and maintaining their positive assessment.
- As suggested by Reviewer DvDX, we have provided details about the optimization process in Appendix Sec.A. We have also conducted an experiment on instance normalization, and discussed more details on component aggregation, decision-making of DVP and reconstruction results. We are glad that our response has addressed the concerns from Reviewer DvDX.
- We have included experiments with complex instructions of spatial relation in Appendix Sec.K, Fig. 18, as suggested by Reviewer EyuE. We have provided clarification regarding the dependency on DVP modules and comparison with other techniques. Furthermore, we have highlighted the importance of the ethical concerns and added discussion in Appendix Sec.N.

In summary, we would like to express our appreciation to Reviewer fWRk, TXue, and DvDx for acknowledging our response. We are particularly grateful that Reviewer fWRk has increased their scores, and Reviewers TXue and DvDx have maintained their positive assessment. Although we understand that Reviewer EyuE has not engaged in subsequent discussion due to their busy schedule, we believe that our response has effectively addressed their concerns through clear explanations and additional illustrative examples.

We would like to emphasize the contributions of our work, which have been acknowledged by the reviewers and are important to the community:
- *Flexible positional editing*: Our system empowers unrestricted editing, including the manipulation of position, thanks to our advanced in-context reasoning capability—a functionality that is not achievable in traditional visual programming. More precisely, we transform human instructions into a domain-specific language, incorporating a comprehensive set of fundamental logic, thereby facilitating versatile image editing.
- *Automatic image translation*: Traditional visual programming relies on off-the-shelf modules that depend on manually crafted guidance scale parameters to oversee the translation process for each individual image, resulting in condition-rigid learning. In contrast, our DVP introduces a novel condition-flexible diffusion model that operates fully automatically, eliminating the need for human intervention. This model outperforms traditional methods by robustly translating images without the need for tunable parameters, representing a substantial advancement in both the quality and versatility of image processing.
- *Quick adaptability*: The DVP framework we present is highly flexible and extensible, adept at addressing a diverse set of tasks that extend beyond the traditional scope. This encompasses, but is not restricted to, video editing, text-to-image generation, etc. The adaptability of our framework is underscored by its compatibility with the off-the-shelf models. This adaptability is largely attributed to our GPT planner, which seamlessly integrates with these models, making DVP easily adaptable to various applications.

Finally, we deeply value the constructive comments provided by the reviewers. In response, we have carefully refined our work based on the feedback received. Considering the contributions made, we hope that DVP can provide new insights to the image translation and visual programming communities, and contribute to their further development.

Sincerely, \
Authors

---

### Meta-Review · Area_Chair_n2XJ · 2023-12-05

**Metareview:**

In this paper, the authors propose Diffusion Visual Programmer (DVP), a neuro-symbolic image translation framework using a diffusion model and a GPT model. In this framework, the authors prompt LLM to generate a sequence of programs which can be executed to plan, segment objects, caption the image, inpaint and manipulate the sizes and positions of objects. Such a framework is quite interesting to study, and is quite useful for interactive editing applications. The authors also show high fidelity synthesis results with good spatial awareness and enhanced user control.

The proposed framework is quite interesting. The experimental results shown are quite strong. The strong controllability achieved by this method is quite appealing.

The novelty improvements compared to other visual programming approaches is somewhat limited. The paper would have been even more stronger with more baselines and a comprehensive analysis.

**Justification For Why Not Higher Score:**

While the contributions provided in this paper are good, I feel the contribution is still somewhat incremental when compared to visual programming approaches.

**Justification For Why Not Lower Score:**

Some of the concerns raised by the reviewers include novelty compared to visual programming approaches and lack of baseline comparisons. Regarding the novelty aspect, I think the paper has non-trivial contributions such as use of instance normalization, arbitrary position manipulations, etc which might be useful to the community. The authors have included some baselines in rebuttal. The paper has also shown good results on sufficiently complex inputs in their paper. Overall, I think the contributions provided in this paper is worthy of publishing.

---

### Decision · Program_Chairs · 2024-01-16

Accept (poster)